# Synapse type-specific proteomic dissection identifies IgSF8 as a hippocampal CA3 microcircuit organizer

Nuno Apóstolo [1,2], Samuel N. Smukowski[3], Jeroen Vanderlinden[1,2], Giuseppe Condomitti[1,2], Vasily Rybakin[4], Jolijn ten Bos [1,2], Laura Trobiani[5], Sybren Portegies[1,2], Kristel M. Vennekens[1,2], Natalia V. Gounko [1,2,6], Davide Comoletti[5,7], Keimpe D. Wierda[1,2], Jeffrey N. Savas [3✉] & Joris de Wit [1,2✉]

Excitatory and inhibitory neurons are connected into microcircuits that generate circuit output. Central in the hippocampal CA3 microcircuit is the mossy fiber (MF) synapse, which provides powerful direct excitatory input and indirect feedforward inhibition to CA3 pyramidal neurons. Here, we dissect its cell-surface protein (CSP) composition to discover novel regulators of MF synaptic connectivity. Proteomic profiling of isolated MF synaptosomes uncovers a rich CSP composition, including many CSPs without synaptic function and several that are uncharacterized. Cell-surface interactome screening identifies IgSF8 as a neuronal receptor enriched in the MF pathway. Presynaptic *Igsf8* deletion impairs MF synaptic architecture and robustly decreases the density of bouton filopodia that provide feedforward inhibition. Consequently, IgSF8 loss impairs excitation/inhibition balance and increases excitability of CA3 pyramidal neurons. Our results provide insight into the CSP landscape and interactome of a specific excitatory synapse and reveal IgSF8 as a critical regulator of CA3 microcircuit connectivity and function.

[1] VIB Center for Brain & Disease Research, Herestraat 49, 3000 Leuven, Belgium. [2] KU Leuven, Department of Neurosciences, Leuven Brain Institute, Herestraat 49, 3000 Leuven, Belgium. [3] Department of Neurology, Northwestern University Feinberg School of Medicine, Chicago, IL 60611, USA. [4] Immunobiology, REGA Institute, Department of Microbiology and Immunology, KU Leuven, Leuven, Belgium. [5] School of Biological Sciences, Victoria University of Wellington, Wellington 6140, New Zealand. [6] Electron Microscopy Platform & VIB BioImaging Core, Herestraat 49, 3000 Leuven, Belgium. [7] Child Health Institute of New Jersey, and Departments of Neuroscience and Cell Biology, Robert Wood Johnson Medical School, Rutgers University, New Brunswick, NJ 08901, USA. ✉email: jeffrey.savas@northwestern.edu; joris.dewit@kuleuven.vib.be

Excitatory and inhibitory neurons are connected into microcircuits and dynamically interact to process information. The specific patterns of connectivity between the neurons in a microcircuit, and the properties and plasticity of transmission at their synapses, are key factors in generating circuit output, from motor to cognitive function[1–3]. Elucidating the molecular mechanisms controlling microcircuit connectivity is important for understanding circuit function in physiology and disease.

The hippocampal mossy fiber (MF) synapse connecting dentate gyrus (DG) granule cells (GCs) and CA3 pyramidal neurons (Fig. 1a) is a central synapse in the CA3 microcircuit that is critical for hippocampal function[4]. A giant MF presynaptic bouton engulfs a postsynaptic element consisting of multiheaded dendritic spines, or thorny excrescences[5–7]. Each MF bouton contains a large vesicle pool and multiple release sites capable of providing powerful excitatory input to CA3 pyramidal neurons. This robust excitation is controlled by strong feedforward inhibition of CA3 pyramidal neurons[8], mediated by filopodia extending from the MF bouton that form excitatory synapses onto interneurons in stratum lucidum (SL)[9] (Fig. 1a). SL interneurons in turn provide inhibition to CA3 pyramidal neurons. As MF-interneuron synapses are more numerous than MF-CA3 synapses[9], this results in net inhibition of CA3 pyramidal neurons. The net inhibition changes to excitation upon an increase in GC firing frequency, due to a rapid depression of the MF-interneuron synapse and a strong facilitation of the MF-CA3 synapse[10,11]. Together, the connectivity and synaptic plasticity properties of the CA3 microcircuit control CA3 pyramidal neuron excitability and firing properties[12,13]. Upon learning, GCs recruit feedforward inhibition through increased connectivity of MF filopodia with fast-spiking SL interneurons, which is important for memory precision[14,15]. Conversely, feedforward inhibition decreases with aging, resulting in CA3 hyperactivity[16]. Precise control of CA3 microcircuit connectivity and function is thus critical for cognitive function.

Cell-surface proteins (CSPs), including transmembrane, membrane-anchored, and secreted proteins, play a key role in the formation of precise connectivity[17–19]. CSPs are expressed in cell type-specific combinations[20–23] and form protein–protein interaction networks that regulate circuit assembly[24–26]. Recent studies reveal a synaptic input-specific localization and function of postsynaptic CSPs in hippocampal pyramidal neuron dendrites[27–30]. Interneuron cell type-specific expression of presynaptic CSPs regulates domain-specific innervation of cortical pyramidal neurons[31]. Together, synaptic input-specific distribution of CSPs in dendrites and differential expression in axonal populations shape circuit connectivity[32]. Comprehensive characterization of the CSP composition of specific connections has remained challenging, however, and the cell-surface interactions regulating CA3 microcircuit connectivity remain poorly understood.

Here, we dissect the CSP composition of the MF synapse to discover novel regulators of synaptic connectivity in the CA3 microcircuit. Using mass spectrometry (MS) to profile the proteome of isolated MF synaptosomes, we identify a rich CSP repertoire that includes adhesion proteins, guidance cue receptors, extracellular matrix (ECM) proteins, and several uncharacterized CSPs. Approximately 80% of these CSPs has not been reported to localize or function at MF synapses and ~50% lacks an annotated synaptic function. Combining proteome and CSP interactome screening with extensive validation identifies IgSF8 as an uncharacterized neuronal receptor strongly enriched in the MF pathway. IgSF8 localizes to both MF bouton and filopodia. Dentate GC-specific deletion of *Igsf8* reduces the number of release sites in MF boutons and robustly decreases the number of filopodia. As a consequence, feedforward inhibition of CA3 pyramidal neurons is strongly reduced and excitability is increased, indicating that IgSF8 is required to maintain excitation/inhibition balance and control excitability in the CA3 microcircuit. Taken together, our results provide insight into the CSP landscape and interactome of a specific excitatory synapse type, uncover multiple uncharacterized synaptic CSPs, and reveal IgSF8 as a regulator of CA3 microcircuit connectivity and function.

## Results

**Isolation of MF synaptosomes.** To isolate MF synapses and start uncovering their CSP composition, we relied on two key features of the MF synapse: its large size and the presence of puncta adherentia (PA), a specialized type of adhesive junction found at large synapses[33] that is morphologically and molecularly distinct from the synaptic junction. We combined two established approaches to take advantage of these features. First, we prepared MF synaptosomes from postnatal (P) day 28 hippocampal homogenate, a time point at which MF synapses have matured[34]. We used a previously established biochemical enrichment method[35] that relies on the large size of MF synapses (Supplementary Fig. 1a), and verified that this method enriches for MF synaptosomes. Electron microscopy (EM) analysis showed the presence of large synaptosomes with a presynaptic compartment packed with synaptic vesicles engulfing a postsynaptic compartment (Supplementary Fig. 1b). To confirm the identity of these synaptosomes, we used the MF synapse markers Synaptoporin (Synpr), a presynaptic vesicle-associated protein[36], and Nectin3, a CA3-enriched (http://mouse.brain-map.org/; https://hipposeq.janelia.org/)[37] adhesion protein that localizes to the postsynaptic side of PA junctions in mature MF synapses[38] (Fig. 1b). As an additional MF synapse marker, we used glutamate receptor ionotropic kainate 5 (GluK5), a predominantly postsynaptically localized kainate receptor enriched at MF synapses[39,40] (Supplementary Fig. 1c). Immunolabeling revealed the presence of all three MF synaptic markers in MF synaptosomes (Supplementary Fig. 1d). Synpr labeling overlapped with the excitatory presynaptic marker vesicular glutamate transporter 1 (VGluT1), whereas Nectin3 labeling was restricted to small puncta (Supplementary Fig. 1d), as expected for a PA-localized postsynaptic protein. GluK5 labeling was similarly restricted to small puncta (Supplementary Fig. 1d). Quantification showed an enrichment of large Synpr- and Nectin3-positive synaptosomes in the MF synaptosome-containing fraction (Supplementary Fig. 1e). We subsequently accelerated the biochemical enrichment procedure by omitting gradient centrifugation and depleting myelin from the sample (Supplementary Fig. 1f). Second, we subjected the biochemical preparation to fluorescence-activated synaptosome sorting[41] to deplete nuclei and further enrich for MF synaptic material. To this end, we live-labeled MF synaptosomes with a fluorophore-conjugated monoclonal antibody against the extracellular domain of Nectin3 (Supplementary Fig. 1g) and with FM4-64 membrane dye to label plasma membrane-bound particles (Fig. 1c and Supplementary Fig. 1h, i), and sorted them in a fluorescent cell sorter.

Western blot (WB) analysis showed the presence of MF synapse markers Synpr and Nectin3 in the sorted MF synaptosome sample, whereas myelin and nuclear markers were largely depleted from the sorted fraction (Fig. 1d). Immunofluorescence analysis confirmed the presence of Synpr-, Nectin3-, and VGluT1-positive large synaptosomes in the sorted sample (Fig. 1e, f). In addition to Synpr-/Nectin3-/VGluT1-positive synaptosomes, VGluT1-positive particles were invariably present (Fig. 1e, f). Such debris has previously been observed following fluorescent synaptosome

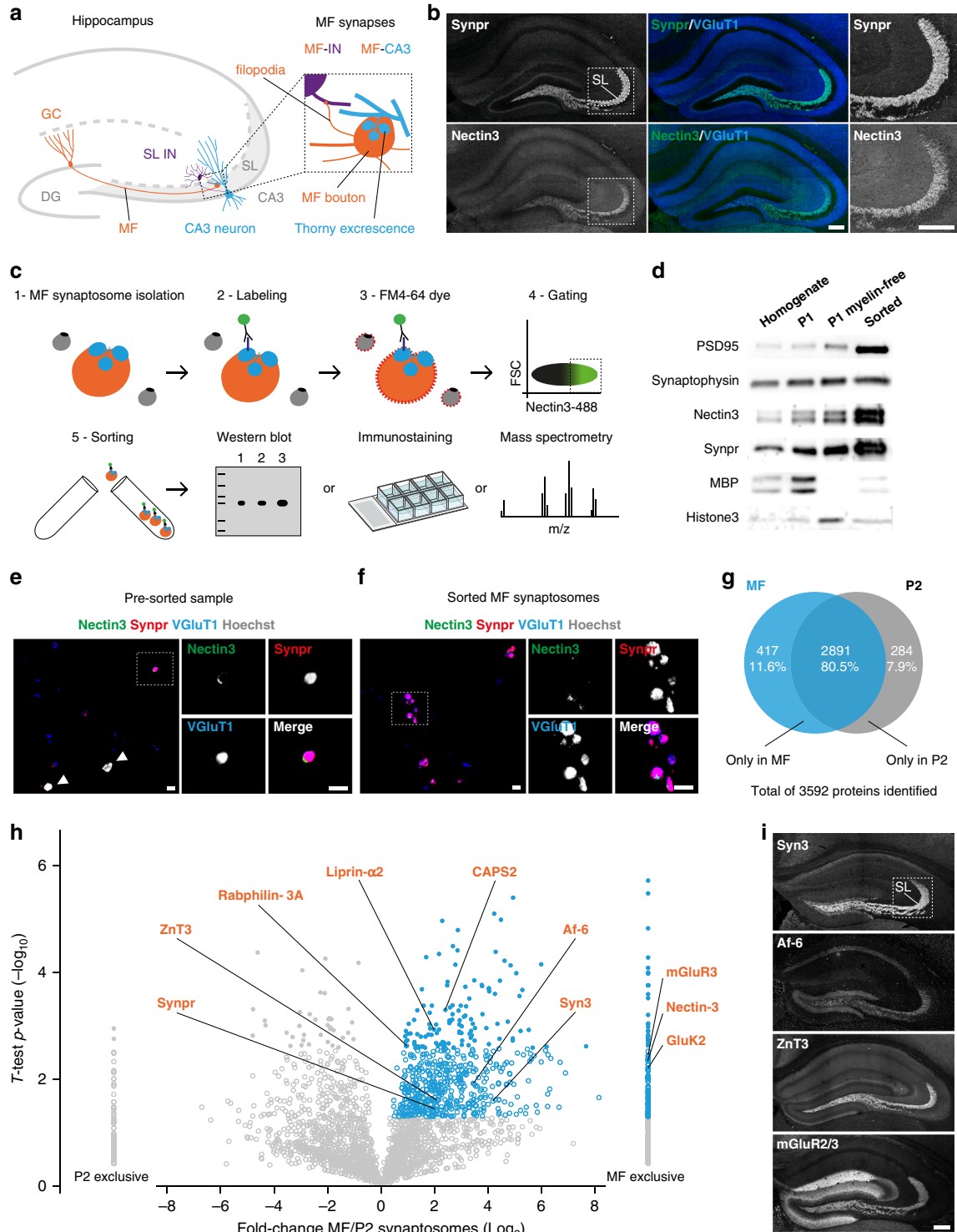

**Fig. 1 Isolation and proteomic profiling of MF synaptosomes. a** Cartoon illustrating the hippocampal MF synapse. GC neurons, mossy fibers (MFs), MF bouton, and filopodia are in orange. CA3 pyramidal neuron and respective thorny excrescences are in blue. Stratum lucidum (SL) interneuron (IN) is in purple. DG dentate gyrus. **b** Confocal images of P28 mouse hippocampal sections immunostained for Synpr, Nectin3, and VGluT1. Magnified insets of the SL in CA3 are shown on the right. **c** Workflow to isolate and analyze MF synaptosomes. **d** Validation of enrichment for MF synaptosomes in sorted material by western blot. MPB, myelin-binding protein. **e**, **f** Confocal images of presorted and sorted material, respectively, immunostained for Nectin3, Synpr, VGluT1, and Hoechst. **g** Venn diagram capturing number and distribution of proteins identified in sorted MF synaptosomes and P2 synaptosomes by LC–MS/MS in three independent experiments (10–12 mice per experiment). **h** Relative distribution of proteins detected in sorted MF synaptosomes and P2 synaptosomes. Significant proteins with positive MF/P2 synaptosome log$_2$ fold change are highlighted in blue ($p$ value ≤ 0.05, two-sided Student's $t$ test). High-confidence measurements at a 5% FDR are shown as closed circles ($q$ value ≤ 0.05, Benjamini–Hochberg correction). A selection of known MF synaptic proteins is annotated in orange. **i** Confocal images of P28 mouse hippocampal sections immunostained for known MF synaptic markers detected in sorted MF synaptosomes. Source data are provided as a Source data file. Scale bars in **b** and **i** 200 μm, in **e** and **f** 5 μm.

sorting[41] and may represent synaptic membrane fragments, resulting from the synaptosome preparation or sorting procedure, indicating that careful validation of potential hits resulting from proteomic analysis of this material is required. Together, these results show that a combination of biochemical enrichment, fluorescent labeling, and sorting successfully enriches for MF synaptosomes from hippocampal tissue homogenate.

**Proteomic profiling of MF synaptosomes**. For proteomic analysis, we compared sorted MF synaptosomes to P2 synaptosomes isolated in parallel from the same hippocampal homogenate (Supplementary Fig. 1f). P2 synaptosomes represent a mixed population of small hippocampal synapses, and serve as a background reference in our analysis. We analyzed sorted MF synaptosomes and P2 synaptosomes from three independent experiments (10–12 mice per experiment) by liquid chromatography tandem MS (LC–MS/MS)-based proteomics. We identified 3592 proteins with at least three peptide identifications among replicates; 11.6 and 7.9% of these proteins were exclusively detected in sorted MF synaptosomes or P2 synaptosomes, respectively (Fig. 1g and Supplementary Data 1). Gene ontology (GO) analysis showed a similar enrichment for synaptic terms in sorted MF and P2 synaptosomes (Supplementary Fig. 1j). We calculated $\log_2$ fold-change enrichment in sorted MF vs P2 synaptosomes using a label-free semiquantitative approach based on the normalized spectral abundance factor (NSAF; Fig. 1h, Supplementary Fig. 1k and Supplementary Data 1). Our analysis revealed 605 proteins significantly enriched in sorted MF synaptosomes and 138 additional significant proteins exclusively detected in MF synaptosomes (Fig. 1h and Supplementary Data 1). This collection comprises multiple proteins previously reported to be strongly enriched at MF synapses, including the synaptic vesicle-associated proteins Synpr, Synapsin3 (Syn3)[42], Rabphilin-3A[43], and zinc transporter 3 (ZnT3/Slc30a3)[44]; the glutamate receptors GluK2/Grik2 (ref. [45]) and metabotropic glutamate receptor 3 (mGluR3/Grm3)[46]; the presynaptic scaffold protein liprin-α2 (Ppfia2)[47]; the dense core vesicle secretion-related protein calcium-dependent secretion activator 2 (CAPS2/Cadps2)[48]; and the PA-associated proteins Nectin3 (listed as Pvrl3 in Supplementary Data 1) and afadin (Af-6/Mllt4)[38] (Fig. 1h and Supplementary Fig. 1l). Using immunohistochemistry (IHC), we confirmed strongly enriched localization of Syn3, Af-6, mGluR2/3, and ZnT3 in SL (Fig. 1i). Relatively few peptides were detected for Nectin3, the marker used to label MF synaptosomes, which may be due to a low abundance and restricted localization of this protein (Supplementary Fig. 1d). Supporting this notion, similar peptide amounts were detected for GluK5/Grik5 (Supplementary Data 1), a transmembrane receptor with a restricted localization in MF synaptosomes (Supplementary Fig. 1d). Together, these results indicate that LC–MS/MS analysis of isolated MF synaptosomes confidently identifies MF synaptic proteins.

**Dissection of MF synapse CSP composition**. Using the UniProt database (https://www.uniprot.org), we next queried our proteomic dataset for transmembrane, membrane-anchored, and secreted proteins among the proteins significantly enriched in sorted MF synaptosomes vs P2 synaptosomes and those exclusively detected in MF synaptosomes. We identified a panel of 77 CSPs, including adhesion proteins, receptors, secreted glycoproteins, receptor protein tyrosine phosphatases, and tyrosine kinases (Fig. 2a and Supplementary Data 2). Most major CSP families, such as the immunoglobulin (Ig) superfamily (IgSF), fibronectin type-III (FN3), and leucine-rich repeat (LRR) family, were represented (Fig. 2b). Only a small proportion (20,8%) of these MF synapse

CSP candidates has been previously reported to localize or function at MF synapses (Supplementary Data 2). GO analysis using the synapse biology SynGO database (https://syngoportal.org)[49] revealed that many of the CSPs identified at MF synapses lack a synaptic function (Fig. 2c). The secreted protein bone morphogenic protein/retinoic acid-inducible neural-specific protein 2 (BRINP2), and the transmembrane receptors family with sequence similarity 171 member A2 (FAM171A2), adipocyte plasma membrane-associated protein (APMAP), and IgSF8 lack a known function in the brain altogether (Fig. 2c).

To validate these results, we tested a large panel of antibodies for detection of CSPs by WB and IHC (Supplementary Fig. 2a and Supplementary Data 2). We validated the presence of 15 CSPs in sorted P28 MF synaptosomes by WB (Fig. 2d). Of the SynGO-annotated CSPs, we confirmed the secreted ECM protein TenR; the transmembrane receptors neuropilin-1 (NRP1), receptor-type tyrosine-protein phosphatase delta (RPTPδ) and sigma (RPTPσ), and Teneurin4, as well as the GPI-anchored receptor contactin 1 (CNTN1; Fig. 2d). Of the CSPs lacking a SynGO-annotated synaptic function, we validated the transmembrane receptors intercellular adhesion molecule 5 (ICAM5), IgSF containing LRR protein 2 (ISLR2), Neogenin, PlexinA1, roundabout homolog 2 (ROBO2), mast/stem cell growth factor receptor SCFR/Kit, and the GPI-anchored receptor neural growth regulator 1 (NEGR1; Fig. 2d). Of the uncharacterized CSPs in the brain, we confirmed the presence of BRINP2 and IgSF8 in sorted MF synaptosomes (Fig. 2d).

Using IHC on P28 mouse hippocampal sections, we validated localization to the MF pathway for 21 CSPs, 15 of which showed a striking laminar distribution (Fig. 2e and Supplementary Fig. 2b). We confirmed strong localization to SL for the SynGO-annotated secreted proteins neuronal pentraxin 1 (NPTX1) and neuronal olfactomedin-related ER localized protein/olfactomedin-1 (OLFM1), as well as the transmembrane receptors neural cell adhesion molecule 1 (NCAM1), NRP1 and NRP2, and LRR and FN3 domain-containing protein 1 (LRFN1/SALM2; Fig. 2e). Of the CSPs without synaptic function, the secreted protein cartilage acidic protein 1 (CRTAC1); the transmembrane receptors NCAM2, ISLR2, Neogenin, ROBO2, and PlexinA3; and the GPI-anchored receptor NEGR1 displayed strongly enriched immunoreactivity in SL (Fig. 2e). Of the uncharacterized CSPs, we validated FAM171A2 in SL, which displayed weak labeling in SL, and IgSF8, which showed strong labeling in the MF pathway (Fig. 2e).

Besides these 15 CSPs with lamina-specific localization patterns, we observed a broad hippocampal distribution, including SL, for an additional six CSPs (Supplementary Fig. 2b). Among these are CNTN1 and TenR, which we also confirmed by WB (Fig. 2d). CSPs detected in sorted MF synaptosomes by WB and localized to SL by IHC were in good agreement (Fig. 2d, e and Supplementary Fig. 2b), with three exceptions: ICAM5, PlexinA1, and SCFR/Kit. While these three CSPs were validated in sorted MF synaptosomes by WB (Fig. 2d), they showed little immunoreactivity in SL (Supplementary Fig. 2c). It is possible that the antibodies used do not recognize the epitopes for these CSPs in SL or that these CSPs are present in MF-sorted synaptosomes, as contaminants and represent false positives. SCFR/Kit, however, has previously been shown to function at MF synapses[50–52] (Supplementary Data 2). In conclusion, dissection of the MF surface proteome reveals a diverse CSP repertoire containing many CSPs that were not previously shown to be associated with synapses and several CSPs of uncharacterized function.

**MF synapse CSP interactome screening**. CSPs exert their function by forming protein–protein interaction networks[17,24].

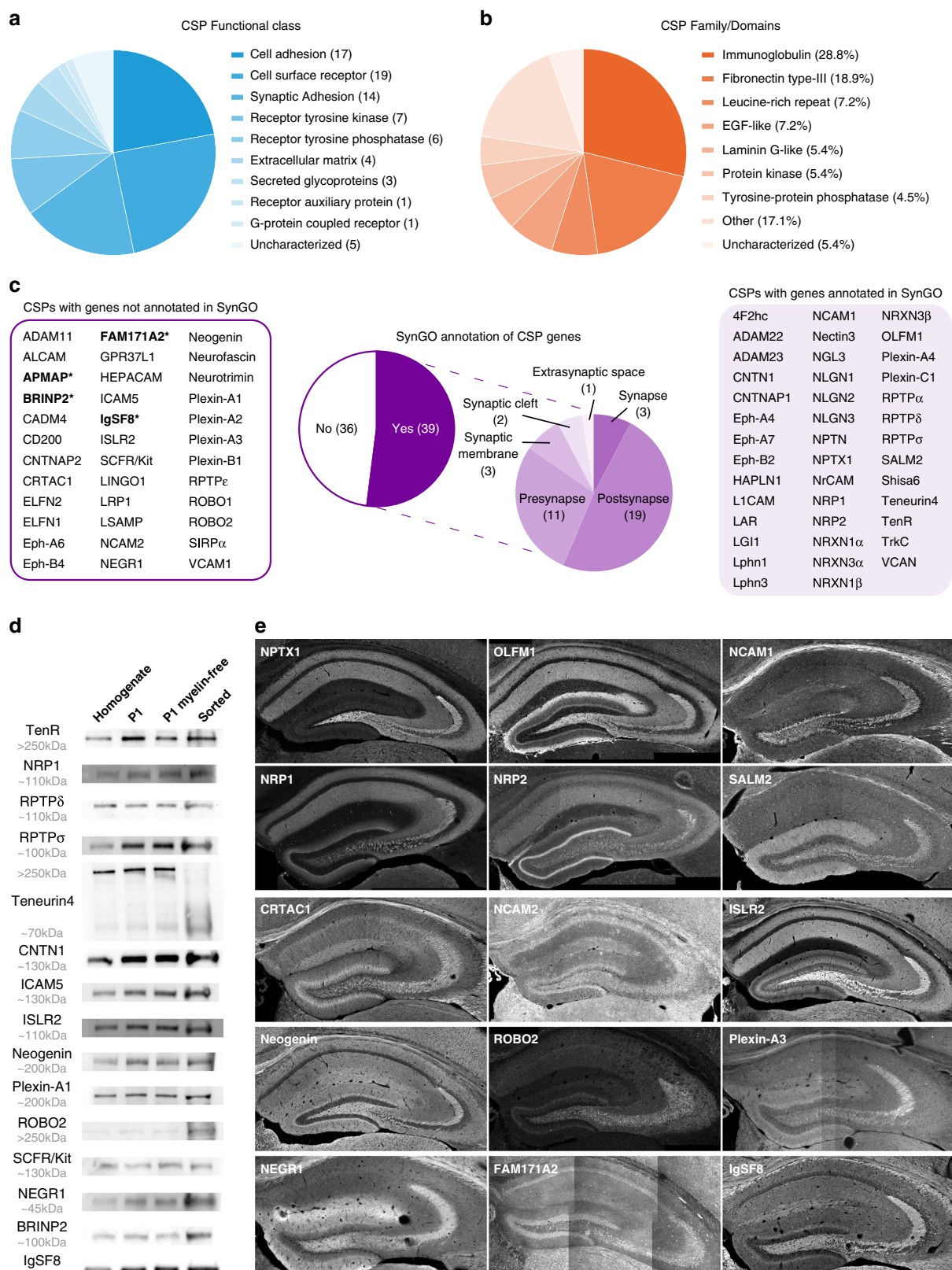

**Fig. 2 Dissection of MF synapse CSP composition. a** Protein functional classes represented in the group of CSPs identified in isolated MF synaptosomes. **b** Occurrence of protein domains among identified CSPs. **c** SynGO cellular component analysis of CSP genes (75) and respective CSPs (77). Discrepancy between gene number and CSP number is related to NRXN isoforms. Proteins with unknown function in the brain are highlighted in bold with an asterisk: APMAP, BRINP2, FAM171A2, and IgSF8. **d** Validation of 15 CSPs in sorted MF synaptosomes by western blot, including BRINP2 and IgSF8, CSPs without a known function in the brain. **e** Confocal images of P28 mouse hippocampal sections immunostained for 15 CSPs showing a striking laminar distribution to the SL. Of the CSPs without a known function in the brain, IgSF8 showed strong labeling in the MF pathway. Source data are provided as a Source data file. Scale bar in **e** 200 μm.

To elucidate ligand–receptor relationships among the CSPs identified in sorted MF synaptosomes and obtain clues about potential roles of the uncharacterized CSPs, we systematically screened for pairwise interactions. We generated a library of constructs containing the extracellular domains of 73 CSPs (see "Methods" section) identified in sorted MF synaptosomes (Fig. 2c and Supplementary Data 2), fused at their C-terminus with human alkaline phosphatase (AP) or the Fc region of human IgG1 (Supplementary Fig. 3a, b). We subsequently generated and validated recombinant AP- and Fc-tagged proteins (Supplementary Data 3 and "Methods" section), and performed a pairwise high-throughput interaction screen using a semiautomated 384-well format enzyme-linked immunosorbent assay (ELISA)-based assay[25,53,54] (Fig. 3a and Supplementary Fig. 3a, b). The screen was designed to test interactions between 73 AP- and Fc-tagged proteins in both orientations (i.e., AP-X vs Fc-Y and AP-Y vs Fc-X), resulting in $73 \times 73 = 5329$ experimental points and screening of 2701 unique pairwise interactions. Positive interactions (blue in Fig. 3a) were defined as those wells that exhibited an $OD_{650} > 5$-fold over background values (FOB) and that were not present in every bait. The vast majority were negative wells, which were either completely clear, or had negligible FOB values (white in Fig. 3a). False-positive wells were characterized by having significant signal in all baits.

The CSP interactome screen reproducibly identified two major modules of known ligand–receptor pairs: the synaptic adhesion molecules neurexins (NRXNs) and neuroligins (NLGNs)[55], and the IgLON IgSF subfamily of synaptic adhesion molecules[56], confirming specificity of the screen. Multiple additional known interactions were reproducibly identified, including Teneurin4 and Latrophilin 3 (Lphn3)[57–59] (Supplementary Fig. 3c and Supplementary Data 3). Combining the results of three independent experiments, using the criteria outlined above and considering only interactions detected at least twice independent of orientation, we identified 38 interacting pairs, 10 of which, to the best of our knowledge, have not been reported before (Fig. 3b and Supplementary Data 3). The interaction of IgSF8 with TenR was among the most reproducible previously unreported ligand–receptor pairs, and IgSF8 was the only uncharacterized CSP for which the interactome screen yielded a binding partner (Fig. 3b and Supplementary Data 3). Thus, CSP interactome analysis identifies multiple ligand–receptor pair modules at MF synapses that may play a role in shaping MF synapse connectivity.

As the MF synapse CSP interactome screen encompassed only those CSPs that reached significance in our proteomic analysis, we next performed affinity chromatography using recombinant IgSF8-ecto-Fc as bait on whole brain synaptosome extract and on MF synaptosome extract to screen a larger panel of prey proteins (Fig. 3c and Supplementary Data 4). Bound proteins were analyzed by LC–MS/MS[60]. We again identified TenR as the main CSP interacting with IgSF8, both in whole brain synaptosome extract, as well as in MF synaptosome extract (Fig. 3d, e and Supplementary Data 4).

To validate the IgSF8–TenR interaction, we used cell-surface binding assays (Fig. 3f) and pull-down assays (Supplementary Fig. 3d) in transfected HEK293T cells, and confirmed the interaction. To test whether IgSF8 and TenR interact directly, we mixed equimolar amounts of Fc control protein, TenR-Fc, or the ECM protein Brevican-Fc, with His-IgSF8 recombinant protein and precipitated Fc proteins. This binding assay showed that the IgSF8–TenR interaction is direct and specific (Fig. 3g). Biolayer interferometry (BLI) using various concentrations of purified IgSF8 and TenR proteins (Supplementary Fig. 3e) determined the dissociation constant ($K_D$) for IgSF8–TenR to be ~1.4 μM (Fig. 3h, i). Taken together, synapse type-specific

proteome dissection combined with CSP interactome screening identifies IgSF8–TenR as a receptor–ligand pair at MF synapses.

**IgSF8 localizes to MF boutons and filopodia.** The identification of TenR, which regulates synaptic structure, plasticity, and excitability[61–63], as an IgSF8 ligand suggests a synaptic function for IgSF8. IgSF8 is a type I membrane protein with a large extracellular domain containing four Ig-like C2-type domains, a transmembrane region, and a short cytoplasmic domain (Fig. 4a). This protein is one of four members of an IgSF subfamily containing the Glu-Trp-Ile (EWI) transmembrane motif of unknown biological function. We first assessed IgSF8's protein expression profile and localization. IgSF8 protein levels in hippocampal lysate mildly increased during postnatal development (Fig. 4b). To determine the synaptic localization of IgSF8, we performed subcellular fractionation, as the conditions required for IgSF8 IHC proved unsuitable for high-resolution imaging. IgSF8 mainly distributed to the Triton-soluble fraction of synaptosomes containing the presynaptic protein synaptophysin (Fig. 4c). In addition, two faint bands were observed in the Triton-insoluble fraction containing the postsynaptic density (PSD) protein PSD95. These biochemical results support a predominantly presynaptic localization of IgSF8, but do not exclude a postsynaptic presence. The IgSF8 antibody was specific, as IgSF8 protein levels were largely abolished in lysate of cultured $Igsf8$ conditional knock out (cKO)[64] hippocampal neurons infected with a lentiviral vector harboring Cre recombinase (Fig. 4d and Supplementary Fig. 4a).

IgSF8 immunohistochemistry of P28 mouse brain sections revealed strong immunoreactivity in CA3 SL and prominent labeling of several fiber tracts, such as corpus callosum (Supplementary Fig. 4b). As $Igsf8$ is strongly expressed in dorsal dentate GCs (https://hipposeq.janelia.org/), we crossed $Igsf8$ cKO mice with the dentate GC-specific $Rbp4$-$Cre$ line (Supplementary Fig. 4a). This largely and selectively abolished IgSF8 immunoreactivity in SL (Fig. 4e), indicating that IgSF8 in the MF pathway is predominantly GC derived. $Igsf8$ cKO did not affect gross MF pathway morphology (Fig. 4e). To gain insight into the subcellular distribution of IgSF8, we first transfected primary cultured neurons. Surface HA-IgSF8 accumulated in distal growth cones (Supplementary Fig. 4c), where it colocalized with phosphorylated Ezrin–Radixin–Moesin (ERM) proteins (Supplementary Fig. 4d), which link cell-surface receptors and the actin cytoskeleton[65], and control neurite outgrowth and process formation[66,67]. We then performed in utero electroporation of embryonic day E15.5 mouse embryos to transfect single dentate GCs, with a plasmid expressing membrane GFP (mGFP) to label MF bouton and filopodia, and HA-tagged IgSF8 to visualize IgSF8 localization at MF synapses (Fig. 4f). HA-IgSF8 localized to both the main MF bouton, as well as to distal filopodia (Fig. 4g and Supplementary Fig. 4e). These results indicate that GC-derived IgSF8 localizes to two key sites in the CA3 microcircuit: the MF bouton synapsing onto CA3 dendrites, and the filopodia synapsing onto SL interneurons.

**Loss of IgSF8 impairs MF synapse architecture and filopodia density.** To determine the consequences of dentate GC-specific loss of IgSF8 on MF synaptic structure, we first analyzed MF bouton ultrastructure in $Igsf8$ cKO mice. We prepared hippocampal sections from P30 $Rbp4$-$Cre:Igsf8$ cKO, and control littermates and imaged synapses by transmission electron microscopy (TEM; Fig. 5a). We observed a clear reduction in active zone (AZ) number and length, and a corresponding decrease in PSD number and length (Fig. 5b), indicating that loss

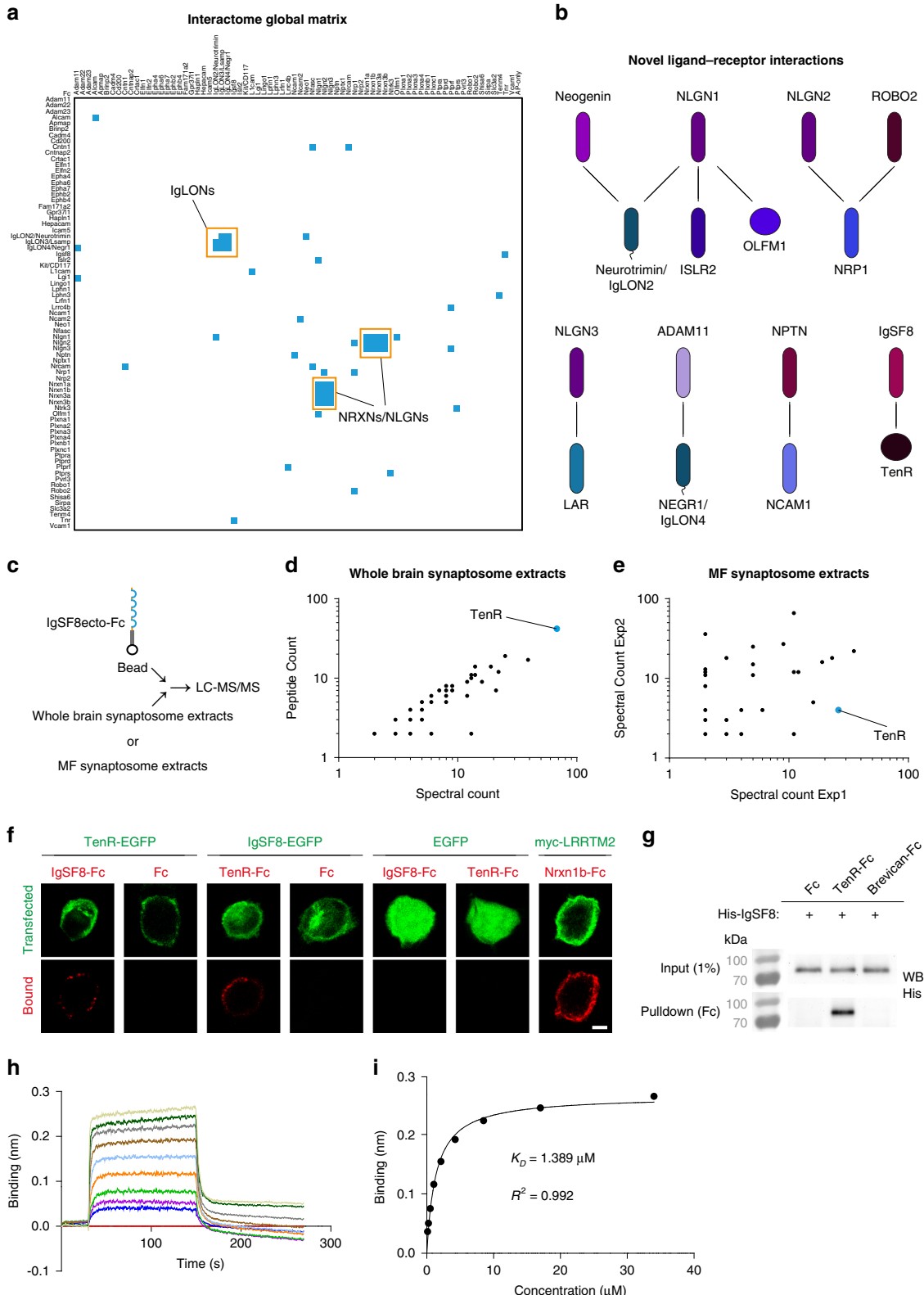

of IgSF8 impairs MF synaptic architecture. MF bouton number and area were not affected in *Igsf8* cKO mice (Fig. 5c, d).

To analyze the morphology of MF bouton and filopodia, we injected adeno-associated viral (AAV) vectors expressing mGFP and Cre recombinase in P7 *Igsf8* cKO mice to remove IgSF8 expression. AAV-mGFP was used as a control (Fig. 5e). We imaged P28 MF boutons using confocal microscopy and

reconstructed them in 3D. We observed a dramatic decrease in the number of filopodia emerging from the main MF bouton in *Igsf8* cKO mice compared to controls (Fig. 5f, g). In addition, we observed a mild reduction in volume of the MF bouton (Fig. 5f, g). Together, these results show that loss of IgSF8 in GCs impairs MF synapse architecture and strongly reduces filopodia density.

**Fig. 3 MF synapse CSP interactome screening. a** Global matrix of data obtained from three independent experiments for interactions between 73 AP- and Fc-tagged proteins in both orientations (i.e., AP-X vs Fc-Y and AP-Y vs Fc-X), resulting in 73 × 73 = 5329 experimental points and 2701 unique pairwise interactions. Columns contain AP-fusion baits, including the AP-only construct as negative control, whereas rows contain Fc-fusion preys. Positive ligand–receptor interactions are indicated in blue. Two major modules of known ligand–receptor pairs are highlighted in orange: the NRXNs–NLGNs and the IgLON family. **b** Interaction networks of novel ligand–receptor pairs identified. Among these, the interaction between IgSF8 and TenR was the only one including a CSP of uncharacterized brain function. **c** Cartoon illustrating the Ecto-Fc MS workflow to validate IgSF8–TenR interaction in synaptosomes. **d** Graph of spectral and peptide counts of proteins captured in a pull-down experiment using whole rat brain synaptosome lysates. Only proteins which were absent in Fc controls and present with ≥2 spectral counts per IgSF8-ecto-Fc experiment are included. TenR is highlighted in blue. **e** Graph of spectral counts of proteins captured in two independent pull-down experiments using P21 mouse MF synaptosome lysates. Only proteins with ≤1 spectral counts in Fc controls and ≥2 spectral counts per IgSF8-ecto-Fc experiment are included. TenR is highlighted in blue. **f** Confocal images of cell-surface binding assays in transfected HEK293T cells show IgSF8–TenR interaction. Negative and positive controls used in the cell-surface binding assays are also shown. **g** Direct binding assays show direct and specific interaction of IgSF8 with TenR, but not with ECM protein Brevican. **h** Biolayer interferometry experiment of the IgSF8–TenR interaction. A concentration range of purified IgSF8 (34–0.133 μM) was used in twofold dilution plus buffer alone (red straight line). **i** Plotting the binding response (nm) vs IgSF8 concentration provided a measure of the affinity of the interaction ($K_D$ ~ 1.4 μM). Source data are provided as a Source data file. Scale bar in **f** 5 μm.

**Loss of IgSF8 impairs spontaneous synaptic transmission in CA3 neurons**. To analyze the functional consequences of IgSF8 removal in GCs on transmission at the MF synapse, we first performed whole-cell voltage-clamp recordings of CA3 pyramidal neurons in acute hippocampal slices from P27–35 *Rbp4-Cre:Igsf8* cKO and control littermates. The frequency of spontaneous excitatory postsynaptic currents (sEPSCs) was strongly reduced in *Igsf8* cKO mice compared to controls (Fig. 6a–c), consistent with the decrease in the number of synaptic junctions at *Igsf8* cKO MF boutons (Fig. 5b). The amplitude of sEPSCs was also reduced in *Igsf8* cKO (Fig. 6d, e), consistent with the reduced length of synaptic junctions (Fig. 5b). Decay time of sEPSCs was not affected (Supplementary Fig. 5a). Histogram analysis showed a loss of large-amplitude sEPSCs in *Igsf8* cKO mice (Supplementary Fig. 5b, c) that originate from MF synapses[68], supporting an impairment of MF-CA3 synapses in the absence of GC-derived IgSF8.

To specifically assess synaptic transmission at MF-CA3 synapses, we injected AAV vectors to express Cre-dependent Channelrhodopsin-2 (DIO-ChR2) in dentate GCs of *Rbp4-Cre: Igsf8* cKO and control littermates, and optically stimulated MF axons while performing whole-cell voltage-clamp recordings from CA3 pyramidal neurons (Fig. 6f). This approach enables us to selectively stimulate MFs lacking IgSF8 in *Igsf8* cKO mice. We analyzed paired-pulse facilitation, a form of short-term plasticity, to assess MF-CA3 presynaptic properties, but found no differences in the amplitude of the first evoked EPSC or the paired-pulse ratio between *Igsf8* cKO and control mice (Fig. 6g–i). Thus, loss of presynaptic IgSF8 impairs spontaneous synaptic transmission, but does not alter evoked transmission at MF-CA3 synapses.

**Reduced feedforward inhibition and increased excitability of CA3 neurons in Igsf8 cKO**. We next assessed the functional consequences of conditional removal of IgSF8 on feedforward inhibition in the CA3 microcircuit. Given the robust decrease in the number of MF filopodia in *Igsf8* cKO mice (Fig. 5g), we hypothesized that stimulation of *Igsf8* cKO MFs would result in reduced recruitment of feedforward inhibition. To test this, we again expressed DIO-ChR2 in dentate GCs of *Rbp4-Cre:Igsf8* cKO and control littermates to optically stimulate MF axons, while performing whole-cell voltage-clamp recordings from CA3 pyramidal neurons (Fig. 7a, b). Light-evoked EPSCs and IPSCs were recorded from the same CA3 pyramidal neuron. We observed a strong reduction in eIPSC amplitude (Fig. 7c) and a robust increase in the eEPSC/eIPSC ratio in *Igsf8* cKO compared to littermate controls (Fig. 7d). Thus, loss of IgSF8 reduces feedforward inhibition in the CA3 microcircuit,

resulting in excitation–inhibition imbalance of CA3 pyramidal neurons.

Feedforward inhibition mediated by MF filopodia plays an important role in the control of CA3 pyramidal neuron excitability[13]. To determine the consequences of reduced feedforward inhibition in *Igsf8* cKO mice on CA3 pyramidal neuron excitability, we optically stimulated MFs and analyzed action potential firing in CA3 pyramidal neurons in *Rbp4-Cre: Igsf8* cKO and control littermates (Fig. 7e). In this experiment, we first recorded CA3 pyramidal neurons in voltage-clamp mode at −82 mV (the Cl⁻ reversal potential) and stimulated MFs to establish a comparable eEPSC amplitude of ±400 pA in both conditions (Fig. 7f). CA3 neurons were then switched to current-clamp mode to record resting membrane potential, which was comparable between both groups (Fig. 7f), followed by optical stimulation of MFs with a 10 Hz train of 20 stimuli. We found that CA3 neurons in *Igsf8* cKO mice started firing action potentials earlier during the train than in control littermates (Fig. 7g), supporting the notion that reduced feedforward inhibition in the absence of IgSF8 leads to increased MF-induced excitability of CA3 neurons. Taken together, these results show that loss of IgSF8 reduces feedforward inhibition resulting in increased excitability of CA3 pyramidal neurons, indicating that IgSF8 is an organizer of hippocampal CA3 microcircuit connectivity and function.

## Discussion

Precise control of the dynamic interaction between excitatory and inhibitory neurons connected into microcircuits is critical for circuit output and cognitive function. Here, we surveyed the proteome of isolated MF synaptosomes to identify novel regulators of synaptic connectivity in the CA3 micro-circuit. Our results uncover a diverse CSP repertoire at MF synapses and reveal the uncharacterized neuronal receptor IgSF8, as a critical regulator of CA3 microcircuit connectivity and function.

**Characterization of the MF synapse CSP landscape**. Dissecting the molecular composition of specific connections remains a major challenge. Sorting of growth cones[69] and specific synapse types (Pfeffer et al.[70], this study), imaging-based approaches[71–73], affinity purification[74], and proximity biotinylation[75–77] each have their advantages and disadvantages[78]. Here, we combined biochemical enrichment, antibody labeling of a synapse type-enriched surface marker, and fluorescent sorting to isolate MF synapses. When coupled with highly sensitive MS analysis, dissecting the proteome of a specific type of synapse with well-characterized connectivity,

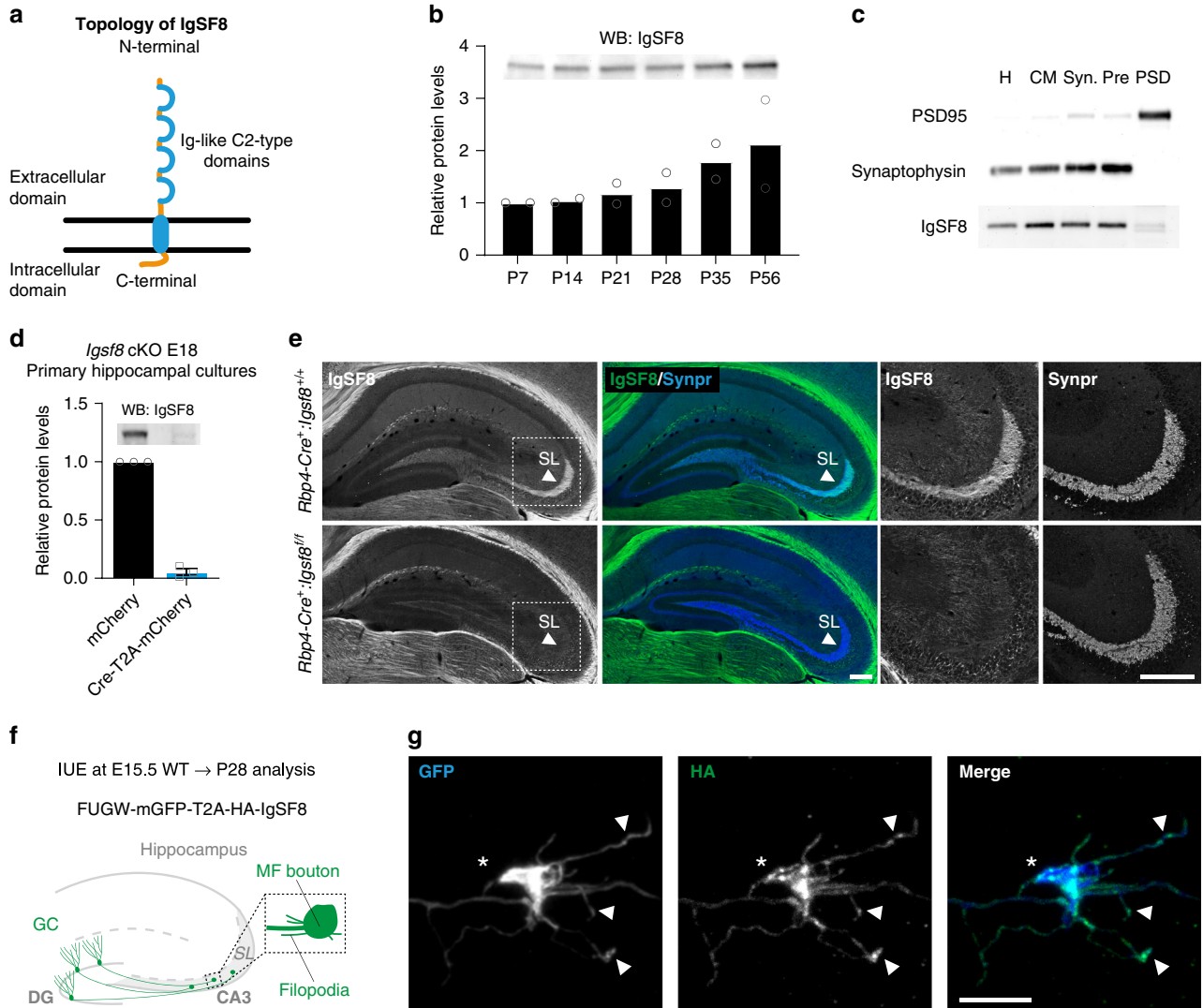

**Fig. 4 IgSF8 localizes to MF boutons and filopodia. a** Cartoon of IgSF8 protein topology. **b** Western blot analysis of IgSF8 protein expression levels in mouse hippocampal homogenates at different developmental time points. Bar graph shows mean. Quantification from two independent experiments. **c** Subcellular fractionation using whole rat brains. H homogenate, CM crude membrane, Syn synaptosomes, Pre presynaptic fraction, PSD postsynaptic density fraction. **d** Western blot analysis of *Igsf8* cKO mouse primary hippocampal cultures infected with a lentiviral vector harboring Cre recombinase or control vector. Bar graph shows mean ± SEM. Quantification from three independent experiments. **e** Confocal images of P28 *Rbp4-Cre:Igsf8* cKO mouse hippocampal sections immunostained for IgSF8 and Synpr. Arrowheads indicate SL. Magnified insets of the SL in CA3 are shown on the right. **f** In utero electroporation setup to sparsely label DG granule cell neurons using membrane GFP (mGFP) and analyze the localization of HA-IgSF8 in MF synapses. **g** Stack of confocal images showing a mGFP-labeled MF bouton and respective filopodia (in blue in the merge). HA-IgSF8 is concentrated in the MF bouton, but also observed in defined regions along the filopodia, including their terminals (in green in the merge). More examples are shown in Supplementary Fig. 4e. Source data are provided as a Source data file. Scale bars in **e** 200 μm and in **g** 10 μm.

morphology, and functional properties, offers the advantage that its molecular composition can be linked to its structural and functional features, at the single-synapse level. Using this approach, we provide the first insight into the CSP landscape and cell-surface interactome of a specific excitatory synapse type.

We identified and validated a rich CSP repertoire at MF synapses. The large majority of the CSPs we identify has not previously been shown to localize or function at MF synapses, and approximately half do not have a synaptic function. Four CSPs (APMAP, FAM171A2, BRINP2, and IgSF8) lack a known function in the brain altogether. Together, our findings underscore the complexity of the MF synaptic proteome and illustrate the advantage of synapse type-specific proteomics in uncovering uncharacterized synaptic proteins.

Our approach identified several CSPs at the MF synapse of uncharacterized function in the brain. The transmembrane protein APMAP interacts with the amyloid precursor protein (APP) and the γ-secretase complex and negatively regulates amyloid-β production[79,80]. The transmembrane protein FAM171A2 induces membrane protrusions in cultured cells upon overexpression and is required for invasive growth of melanoma cells[81]. Immunohistochemistry validated the presence of FAM171A2 protein in the MF pathway, but its role there remains unknown. The secreted glycoprotein BRINP2 is highly expressed in CA3 (ref. [82]) and validated to be present in MF synaptosomes. *Brinp2* KO mice display hyperactivity[83], but the function of BRINP2 is not known. Finally, the transmembrane protein IgSF8 interacts with tetraspanins and integrins[84–86], and is thought to modulate cell motility

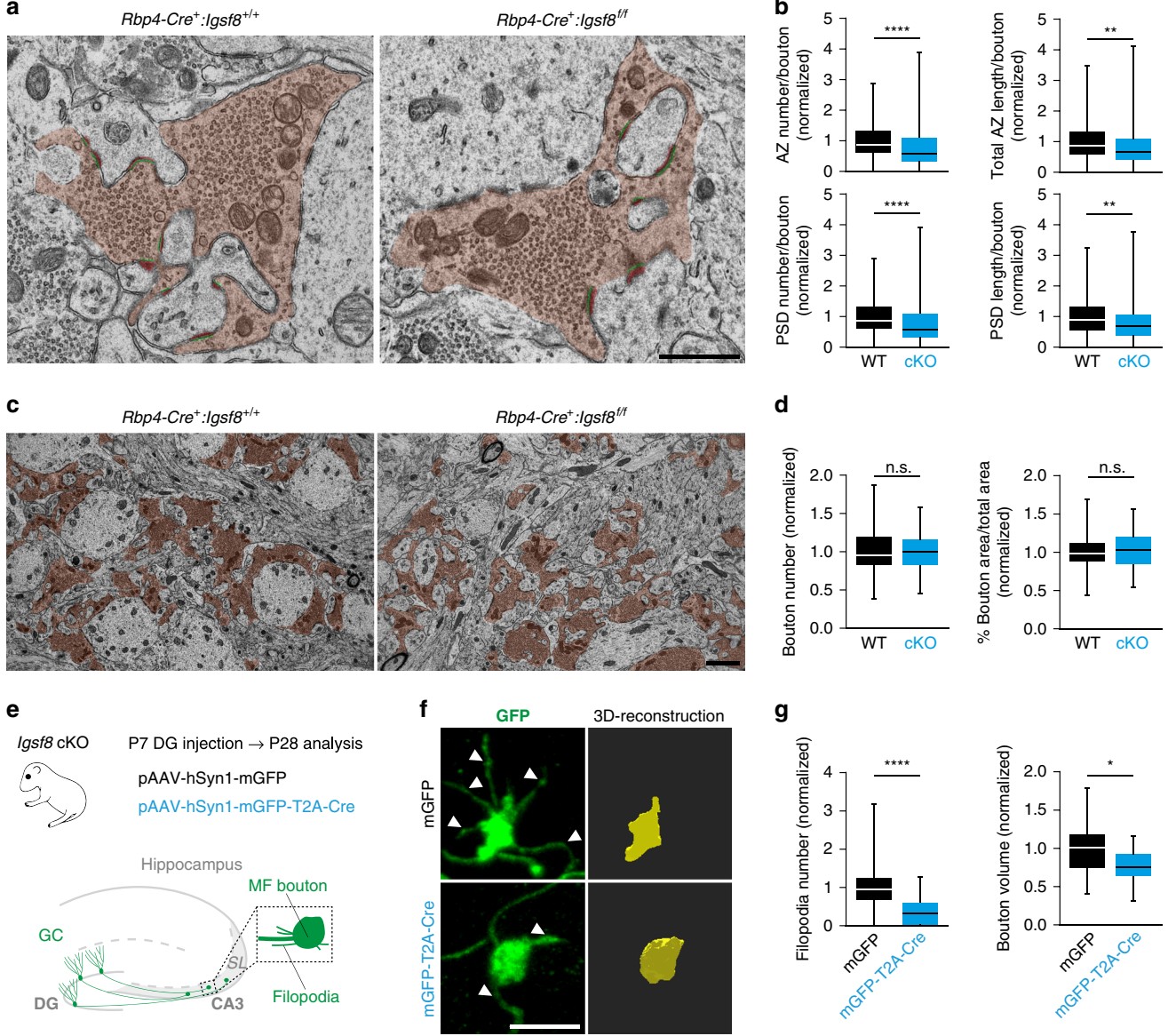

**Fig. 5 Loss of IgSF8 impairs MF synapse architecture and filopodia density. a** Electron microscope images of MF-CA3 synapses from *Rbp4-Cre:Igsf8* cKO and WT littermates (5000× magnification). MF boutons are highlighted in orange. AZs and PSDs are highlighted in green and red, respectively. **b** Graphs show quantification of analysis done in **a** in littermate mice examined over three independent experiments (WT, *n* = 135 boutons and cKO, *n* = 136). ****P* < 0.0001 and ***P* = 0.0028 in upper panel. *****P* < 0.0001 and ***P* = 0.0056 in lower panel. **c** Electron microscope images of MF-CA3 synapses from *Rbp4-Cre:Igsf8* cKO and WT littermates (2500× magnification) to analyze number and area of MF boutons, highlighted in orange. **d** Graphs show quantification of analysis done in **c** in littermate mice examined over three independent experiments (WT, *n* = 103 images and cKO, *n* = 93). **e** Experimental design to analyze structural changes in MF synapses following deletion of *Igsf8* specifically in DG granule cells. **f** Stacks of confocal images of individual MF boutons (left) and respective 3D reconstructions (right) to analyze number of MF bouton filopodia and MF bouton volume. Arrowheads show filopodia emerging from MF boutons. **g** Graphs show quantification of analysis done in **f** in littermate mice examined over three independent experiments (mGFP, *n* = 27 boutons and mGFP-T2A-Cre *n* = 29). *****P* < 0.0001 and **P* = 0.0149 in **g**. Box-and-whisker plots in **b**, **d** and **g** show median, interquartile range, minimum, and maximum. Two-sided Mann–Whitney tests were used in **b** and **g**. Unpaired Student's *t* tests were used in **d**. n.s. not significant. Source data are provided as a Source data file. Scale bars in **a** 1 μm, in **c** 2 μm, and in **f** 5 μm.

in immune and cancer cells through tetraspanin microdomains[87,88]. Interestingly, both FAM171A2 and IgSF8 are involved in membrane protrusion and motility, which might be relevant for structural plasticity of MF synapses[89].

Multiple secreted and ECM proteins were validated at MF synapses (BRINP2, CRTAC1, NPTX1, OLFM1, and TenR). The synaptic function of CRTAC1, which acts as a Nogo receptor-1 antagonist in regulating lateral olfactory axon tract bundling in early development[90], is not known. Many CSPs with roles in the early stages of circuit assembly (NRP1, NRP2, CRTAC1, ISLR2,

Neogenin, ROBO2, PlexinA1, and PlexinA3) were validated at mature MF synapses. With the exception of the NRPs[91–93], their synaptic role remains unknown. Our findings suggest that secreted/ECM proteins and axon guidance-related proteins shape MF synaptic connectivity and continue to play a role at mature MF synapses.

Several CSPs with an established role in MF synapse development and function, including the postsynaptic receptors EphB2 (ref. [94]) and EphA4 (ref. [95]), and the presynaptic adhesion molecule NRXN1 (ref. [96]), were identified (see Supplementary Data 2 for a

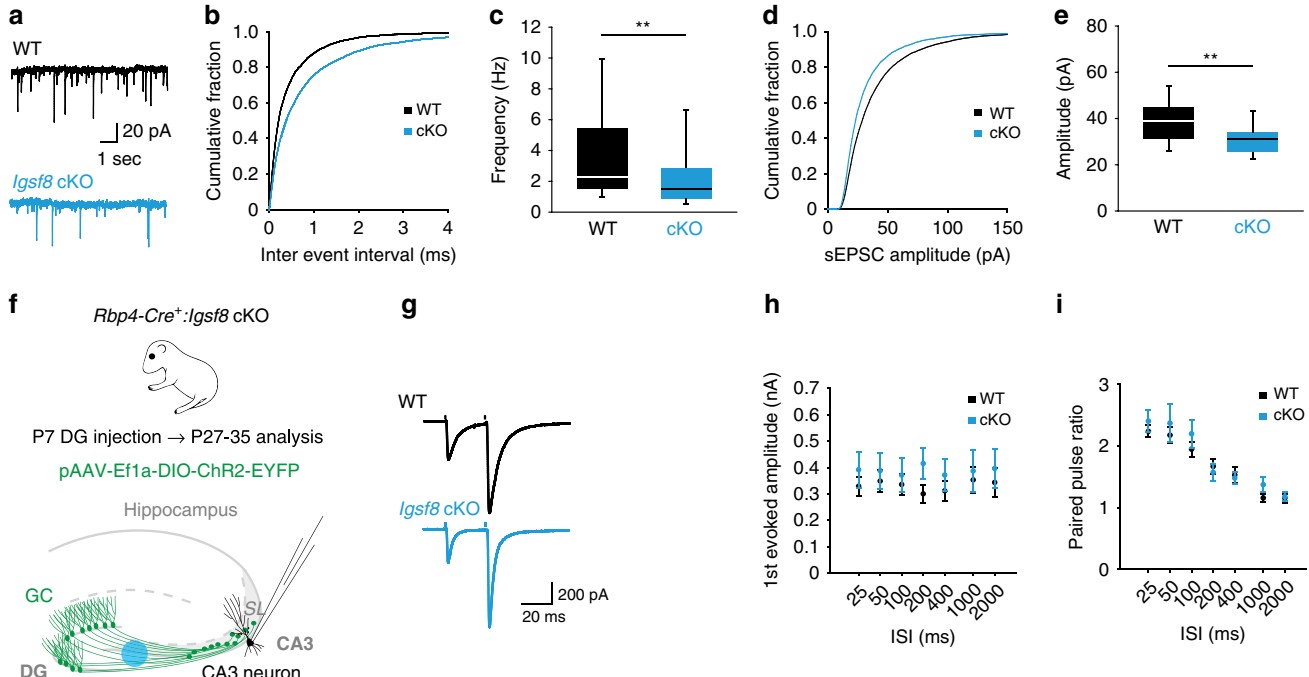

**Fig. 6 Loss of IgSF8 impairs spontaneous synaptic transmission in CA3 neurons. a** Representative sEPSC traces from whole-cell voltage-clamp recordings of CA3 neurons in acute hippocampal slices of *Rbp4-Cre:Igsf8* cKO and WT littermates. **b** Cumulative distribution of sEPSC inter-event intervals. **c** Quantification of sEPSC frequency in littermate mice examined over three independent experiments (WT, $n = 31$ neurons and cKO, $n = 38$). $**P = 0.008$. **d** Cumulative distribution of sEPSC amplitudes. **e** Quantification of sEPSC amplitudes in littermate mice examined over three independent experiments (WT, $n = 31$ neurons and cKO, $n = 38$). $**P = 0.002$. **f** Cartoon illustrating whole-cell voltage-clamp recordings of CA3 neurons to measure MF-evoked responses in acute hippocampal slices of *Rbp4-Cre:Igsf8* cKO and WT littermates using optogenetics. **g** Representative traces of paired-pulse ratio in *Rbp4-Cre:Igsf8* cKO and WT littermates using optogenetics. **h** Quantification of first evoked amplitudes in littermate mice examined over three independent experiments (WT, $n = 23$ neurons and cKO, $n = 21$). **i** Quantification of paired-pulse ratios in littermate mice examined over three independent experiments (WT, $n = 23$ neurons and cKO, $n = 21$). Box-and-whisker plots in **c** and **e** show median, interquartile range, minimum, and maximum. Graphs in **h** and **i** show mean ± SEM. Two-sided Mann–Whitney tests were used in **c** and **e**. Source data are provided as a Source data file.

complete overview). In addition, the secreted glycoprotein C1QL3 (ref. [97]), the classic type II cadherin Cadherin-9 (ref. [36]), and the G-protein-coupled receptor-like protein GPR158 (ref. [27]) were also detected in sorted MF synaptosomes (Supplementary Data 1). Interestingly, the loss of function phenotypes of these proteins display similarities, but are rarely identical to one another. Together, these findings suggest that a combinatorial code of CSPs that act in a partially redundant manner defines MF synapse identity. The CSP diversity at the MF synapse may also reflect heterogeneity within this synaptic population. The maturational state of MF synapses varies due to the continuous integration of newborn GCs into the hippocampal circuit[98]. Different histories of synaptic activity may also diversify CSP composition.

**Characterization of the MF synapse cell-surface interactome.** We used automated interactome screening to map ligand–receptor relationships among the MF synapse CSPs we identified, characterizing a synapse type-specific network of CSP modules and uncovering ten interactions that have not been reported previously. Nine of these remain to be validated by independent methods, but several are worth highlighting here. The secreted glycoprotein OLFM1 interacts with APP[99] and AMPARs[100–103], and regulates the mobility of AMPARs and short-term plasticity in cultured neurons[104]. The OLFM1–NLGN1 interaction identified here suggests that OLFM1 is well positioned to modulate excitatory neurotransmission and postsynaptic plasticity through different protein complexes. We further identified several ligand–receptor modules featuring IgLON family members, including the interaction of

Neurotrimin/IgLON2 with NLGN1 and of NEGR1/IgLON4, with the adhesion molecule ADAM11. IgLONs are synaptic adhesion molecules that span the synaptic cleft[56]. NEGR1/IgLON4 regulates synapse number in cultured hippocampal neurons[105]. Future studies addressing the synaptic functions of IgLONs should take the putative binding partners identified here into consideration. Finally, we identified several interactions, including those of NLGN1–ISLR2, NLGN2–NRP1, and Neurotrimin/IgLON2-Neogenin, that suggest considerable cross talk between synaptic adhesion molecules and axon guidance-related receptors.

As the MF synapse cell-surface interactome screen was limited to those CSPs that reached significance in our proteome analysis, tests only binary interactions, does not take splice variants into account, and does not detect binding affinities weaker than ±10 μM (refs. [25,26,53,56]), the MF synapse CSP interactome will likely be even more complex. Synapse type-specific proteome analysis and cell-surface interactome analysis will be used in future studies to determine to what extent developmental stage, and neural activity influence cell-surface composition and interactome of MF synapses.

**IgSF8 regulates CA3 microcircuit connectivity and function.** Synapse type-specific proteome profiling and CSP interactome screening uncovered IgSF8 as an uncharacterized neuronal receptor for the ECM protein TenR at the MF synapse. A previous immunohistochemical study demonstrated transient IgSF8 immunoreactivity in olfactory sensory neuron axon terminals during synaptogenesis[106]. Lesion-induced reformation of

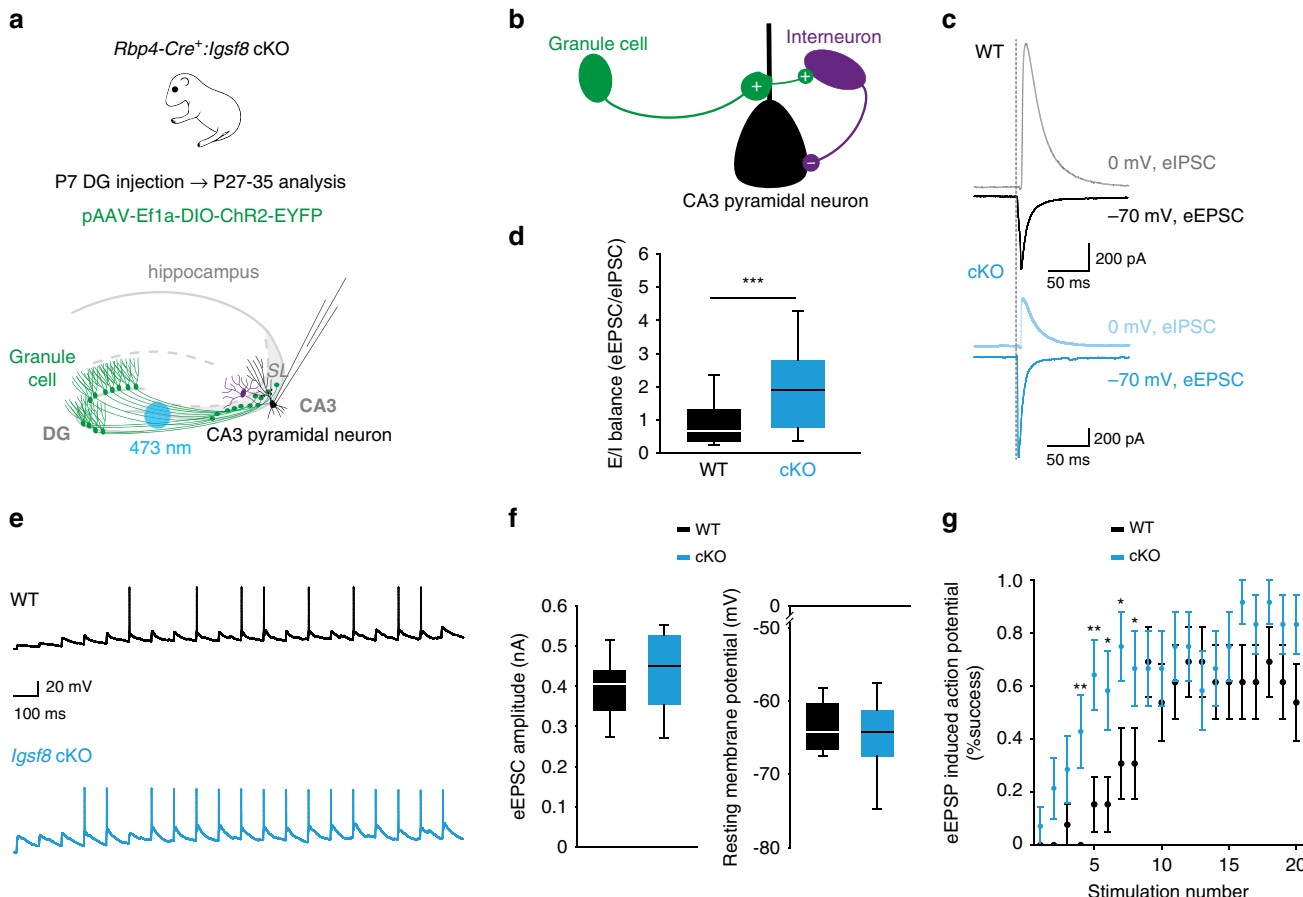

**Fig. 7 Reduced feedforward inhibition and increased excitability of CA3 neurons in *Igsf8* cKO. a** Cartoon illustrating whole-cell voltage-clamp recordings of CA3 neurons to measure MF-evoked responses in acute hippocampal slices of *Rbp4-Cre:Igsf8* cKO and WT littermates using optogenetics. **b** Cartoon illustrating feedforward inhibition microcircuit in CA3. Plus and minus signs represent excitatory and inhibitory synapses, respectively. **c** Representative eEPSC and eIPSC traces from CA3 pyramidal neurons in WT or cKO mice. **d** Quantification of excitation–inhibition balance in CA3 neurons in WT and cKO mice in littermate mice examined over three independent experiments (WT, $n = 35$ neurons and cKO, $n = 29$). ***$P = 0.0003$. **e** Representative eEPSP traces from CA3 pyramidal neurons in WT or cKO mice. **f** Quantification of resting membrane potentials and eEPSC amplitudes of CA3 neurons in WT and cKO littermate mice examined over three independent experiments (WT, $n = 13$ neurons and cKO, $n = 14$). **g** Quantification of induced action potential firing in CA3 neurons in WT and cKO littermate mice examined over three independent experiments (WT, $n = 13$ neurons and cKO, $n = 14$) after a 10 Hz train of 20 stimuli. *$P = 0.03$ and **$P = 0.01$ in **g**. Box-and-whisker plots in **d** and **f** show median, interquartile range, minimum, and maximum. Graph in **g** shows mean ± SEM. Two-sided Mann–Whitney tests were used in **d** and **g**. Source data are provided as a Source data file.

synapses caused IgSF8 to reappear in terminals, whereas blocking olfactory activity prevented the disappearance of IgSF8 from mature synapses[106]. Similarly, the EWI-subfamily member IgSF3 transiently localizes to developing cerebellar GC axon terminals[107]. The localization of IgSF8 to the MF bouton and filopodia suggested a functional requirement for IgSF8 at both sites. Conditional deletion of *Igsf8* in dentate GCs reduced length, and number of AZs and PSDs in the main MF bouton, suggesting that IgSF8 regulates synaptic junction architecture and number. This defect is reminiscent of the reduction in AZ length and number at perisomatic inhibitory synapses in CA1 pyramidal neurons in *Tnr* KO mice[62]. Together, these observations suggest that IgSF8 regulates synapse number and length, possibly via an interaction with TenR, which localizes to the excitatory synaptic cleft of cultured neurons[108].

In addition to defects in the main MF bouton, conditional deletion of *Igsf8* in dentate GCs strongly decreased the number of MF filopodia that synapse onto SL interneurons[9]. The cytoplasmic tail of IgSF8 binds ERM proteins[109] and alpha-actinin[110], both of which are linked to filopodia formation. In cultured neurons, we observed that cell-surface IgSF8 colocalized with phosphorylated (activated) ERM proteins. Activation of

ERM proteins in growth cone filopodia is required for axon growth[111,112]. Treating cultured chick tectal neurons with TenR induces the formation of actin-rich filopodia[113]. Together, these observations suggest that IgSF8, via an interaction with TenR, may promote the formation and/or stabilization of MF filopodia.

MF filopodia mediate feedforward inhibition of CA3 neurons, which plays a key role in CA3 microcircuit function in normal and pathological conditions[12,14–16]. Interestingly, TenR immunoreactivity is highly concentrated around parvalbumin-positive SL interneurons[61,63], suggesting that the interaction of IgSF8 and TenR may control filopodia-SL interneuron connectivity, and play an important role in controlling dynamic interactions between excitatory and inhibitory neurons in the CA3 microcircuit that are important for memory. Although speculative, the localization of IgSF8 to both the main MF bouton and filopodia might constitute a molecular mechanism to coordinate changes at both types of synapses in the CA3 microcircuit in response to changes in activity.

In conclusion, using proteomic profiling of a specific synapse in the CA3 microcircuit, we identified multiple uncharacterized synaptic CSPs and uncovered IgSF8 as a regulator of connectivity within this circuit. Although our study focused on the cell-surface

proteome of the MF synapse, our dataset can be used to analyze other classes of proteins important for synaptic function and may be of use in efforts to model synaptic connectivity in the DG[114]. A combination of approaches, including the synapse type-specific proteomic profiling approach we apply here, will ultimately be required to elucidate the molecular mechanisms underlying the highly precise patterns of connectivity that are required for proper circuit function.

## Methods

**Animals**. All animal experiments were conducted according to the KU Leuven ethical guidelines and approved by the KU Leuven Ethical Committee for Animal Experimentation (approved protocol numbers ECD P037/2016, P014/2017, and P062/2017). Mice were maintained in a specific pathogen-free facility under standard housing conditions with continuous access to food and water. Mice used in the study were 1–8-weeks old and were maintained on 14 h light, 10 h dark light cycle from 7 to 21 h. Wild-type (WT) C57BL/6J mice were obtained from JAX. The *Igsf8* cKO mouse line was obtained from the Riken Institute[64], while the *Rpb4-Cre* line was obtained from GENSAT. Genotypes were regularly checked by PCR analysis. For euthanasia, animals were either anesthetized with isoflurane, and decapitated or injected with an irreversible dose of ketamine–xylazine.

**Neuronal cultures**. Hippocampal neurons were cultured from E18 *Igsf8* cKO mice. Neurons were plated on poly-D-lysine (Millipore) and laminin (Invitrogen)-coated glass coverslips (Glaswarenfabrik Karl Hecht). Neurons were maintained in Neurobasal medium (Invitrogen) supplemented with B27, glucose, glutamax, penicillin/streptomycin (Invitrogen), 25 μm β-mercaptoethanol (Sigma-Aldrich), and insulin (Sigma-Aldrich). To knock out *Igsf8*, neurons were infected with low-titer lentivirus expressing mCherry or mCherry-T2A-Cre 2 days after plating and collected at day in vitro 10 (DIV10).

Cortical neurons from E18 WT mice were cultured and maintained over an astroglial feeder layer in 60-mm culture dishes containing Neurobasal medium. Neurons grew face down over the feeder layer, but were kept separate from the glia by wax dots on the neuronal side of the coverslips. To prevent overgrowth of glia, neuron cultures were treated with 10 μM 5-Fluoro-2′-deoxyuridine (Sigma-Aldrich) at DIV3. Cultures were maintained in a humidified incubator of 5% (vol/vol) $CO_2$/95% (vol/vol) air at 37 °C, feeding the cells once per week by replacing one-third of the medium in the dish[115]. Neurons were electroporated just before plating using an AMAXA Nucleofector kit (Lonza, Basel, Switzerland) or transfected with calcium phosphate at DIV1 with plasmid FUGW-GFP-T2A-HA-IgSF8. At DIV7/8, neurons were fixed and immunostained using the primary antibodies rat anti-HA (1:500; Roche) and rabbit anti-pERM (1:1000; Cell Signaling) with standard procedures. To stain the pool of HA-IgSF8 at the surface, primary antibody anti-rat HA was incubated after fixing neurons, but before the permeabilization step. Anti-pERM primary antibody was incubated after neurons were permeabilized.

**Cell Lines**. HEK293T-17 human embryonic kidney cells (available source material information: fetus) were obtained from American Type Culture Collection (ATCC) cat# CRL-11268. HEK293T-17 cells were grown in Dulbecco's modified Eagle's medium (DMEM; Invitrogen) supplemented with 10% fetal bovine serum (FBS; Invitrogen) and penicillin/streptomycin (Invitrogen).

**Plasmids**. For the ELISA-based interactome assay, we used recombinant AP- or Fc-tagged constructs containing human or mouse extracellular domain (ectodomain) protein sequences derived from the appropriate entry in the UNIPROT database (Uniprot.org). Cloning boundaries and cloning strategy can be found elsewhere[53,56]. Briefly, DNA sequences encoding the ectodomains of transmembrane and GPI-anchored CSPs, or the entire protein sequence in the case of secreted proteins, lacking the signal peptide, were cloned in frame using the 5′ NotI and 3′ XbaI restriction sites of a modified pCMV6-XL4 expression vector. Synthesis of DNA sequences and cloning into pCMV6-XL4 were outsourced. Fc-fusion proteins contain a leader peptide (PLP, prolactin leader peptide) followed by a N-terminal FLAG tag, ectodomain of interest, a 3CPro cleavage site, and the dimeric human Fc domain. Similarly, AP-fusion proteins contain a leader peptide, a FLAG tag and ectodomain of interest. C-terminal to the ectodomain there is a human AP followed by a His6 tag. For cloning NRXN ectodomains, we used DNA sequences of rat NRXN1α−(-S4), mouse NRXN1β−(-S4), mouse NRXN3α−(-S4) (cDNA gift from Ann Marie Craig's lab), and rat NRXN3β−(-S4) (cDNA from Addgene plasmid #58269 from Peter Scheiffele's lab). All constructs were verified by DNA sequencing. The fact that AP is enzymatically active provides a simple way to test expression and measure accurate protein concentration. The 146 recombinant AP- or Fc-tagged constructs used for this ELISA interaction screen were secreted into the media after transient transfection, and used in the assay without further purification. Four CSPs were not included in the ELISA-based interactome assay: CNTN1AP is not expressed in the absence of CNTN1; LRP1 and VCAN are very

large proteins, and costly to synthesize; and RPTPε has a very small ectodomain making potential interactions difficult to detect using this assay.

Full-length cDNAs encoding mouse IgSF8 (BC048387) and Tenascin-R (BC138043) were purchased from Origene and Source Bioscience, respectively. Both cDNAs were cloned into the pEGFP-N1 vector (Clontech) between restriction sites HindIII and SalI to place a C-terminal EGFP tag. Full-length cDNA encoding mouse Brevican (BC052032), lacking the signal peptide, was purchased from Source Bioscience and cloned into the Fc-tagged construct between restriction sites NotI and SalI.

To generate FUGW-mGFP-T2A-HA-IgSF8 two DNA fragments were PCR-amplified with overlapping sequences, full-length cDNA encoding IgSF8 (BC048387) with an additional HA tag, and mGFP, including T2A and HA sequences, and inserted in the FUGW-GFP vector (Addgene #14883) between restriction sites BamHI and EcoRI. To generate FUGW-GFP-T2A-HA-IgSF8, two DNA fragments were PCR-amplified with overlapping sequences, IgSF8 (BC048387) with additional HA tag and T2A sequences, and cytosolic GFP, including a T2A sequence, and inserted in the FUGW-GFP vector (Addgene # 14883) between restriction sites BamHI and EcoRI.

To generate pAAV-hsyn1-mGFP-T2A-Cre, EGFP-T2A-Cre was amplified by PCR from pRetroX-GFP-T2A-Cre (Addgene #63704) and inserted between restriction sites BamHI and HindIII of pAAV-hSyn1-ChR2-EGFP (Addgene #58881). After, was PCR-amplified from FUGW-mGFP-T2A-HA-IgSF8 and replaced by EGFP in pAAV-hsyn1-EGFP-T2A-Cre. As a control vector, pAAV-hsyn1-mGFP was generated by inserting PCR-amplified mGFP between restriction sites BamHI and HindIII of pAAV-hSyn1-ChR2-EGFP (Addgene #58881). The plasmid pAAV-Ef1a-DIO-hChR2(E123T/T159C)-EYFP was obtained from Addgene (#35509). A complete list of primers used to clone these constructs is available in Supplementary Table 1.

**Isolation of MF synaptosomes**. For initial characterization and comparison of crude MF synaptosomes and standard hippocampal synaptosomes, we followed the protocol as described in Taupin et al.[35]. Briefly, hippocampi were dissected quickly in ice-cold Hank's Balanced Salt Solution (HBSS) and homogenized in homogenization buffer (0.32 M sucrose, 5 mM Trizma Base, 1 mM MgCl₂, pH 7.4) with protease inhibitors (pepstatin A, leupeptin, aprotinin, and PMSF) using a Dounce homogenizer. Homogenate was spun at $1000 \times g$ to pellet MF synaptosomes with nuclei and large cell debris in pellet 1 (P1), while general small synaptosomes are in the supernatant (S1). P1 and S1 fractions were loaded on Percoll gradients (Thermo-Fisher), and spun at $32,500 \times g$ at 4 °C for 7 min to separate MF synaptosomes and standard synaptosomes from myelin, nuclei, or mitochondria, resulting in PI (crude MF) and SI (small hippocampal) synaptosome fractions. To prepare MF synaptosomes for sorting, we adapted the protocol of Taupin et al. using 10–12 postnatal day 28 (P28) WT mice per experiment. Hippocampi were dissected quickly in ice-cold HBSS and homogenized in homogenization buffer (0.32 M sucrose, 4 mM Hepes, 1 mM MgCl₂, pH 7.4) with protease inhibitors (pepstatin A, leupeptin, aprotinin, and PMSF) using a Dounce homogenizer. After, the homogenate was filtered through a series of cell strainers (100, 70, and 30 μm) and spun at $1000 \times g$ for 10 min at 4 °C to prepare P1, which includes large MF synaptosomes, and S1. P1 was washed once by resuspension in homogenization buffer and recentrifuged as above. Supernatants S1 were pooled and spun at $15,000 \times g$ for 20 min at 4 °C to prepare postnuclear pellet (P2) synaptosomes. P1 and P2 were resuspended in PBS and myelin was depleted from both fractions according to the instructions of a commercial myelin-removal kit (Miltenyi Biotec), before sorting MF synaptosomes from myelin-depleted P1.

**Fluorescence-activated synaptosome sorting**. Myelin-depleted MF synaptosomes were prepared as described above. After, MF synaptosomes were labeled in non-permeabilizing conditions in PBS with an anti-Nectin3 monoclonal antibody (1:50; Hycult Biotech) directly conjugated with CF488A fluorophore (Sigma-Aldrich) after depletion of bovine serum albumin (BSA) present in the antibody storage solution, using an antibody clean-up kit (Thermo-Fisher). Immediately before running the samples in the cytometer, FM4-64 (1:400; Thermo-Fisher) was added to myelin-depleted MF synaptosomes in order to label all membrane-containing material. FM4-64/Nectin3-488-double-positive MF synaptosomes were sorted on a BD FACS Aria III using an 85 μm nozzle at 45 p.s.i. The pressure differential was set to two to minimize sheer stress. Nectin3-488 was excited using a 488 nm 13 mW laser and detected with a 530/30 band-pass filter. FM4-64 was excited using a 561 nm 30 mW laser and detected with a 720/40 band-pass filter. Unstained and single-stained CF488A and FM4-64 synaptosomes were analyzed to calculate autofluorescence and signal above background, respectively, and Nectin3-488 and FM4-64 gates were set accordingly. Synaptosomes were identified by back-gating on fluorescence using FSC and SSC, with SSC set on a five decade log scale. FM4-64/Nectin3-488-double-positive MF synaptosomes were analyzed on the BD FACS Aria III post sort to quantify the purity of the sorted population.

**Protein expression**. For the ELISA-based assay, proteins were expressed in Expi293 suspension-adapted cells (ThermoFisher Scientific). These cells were cultured in Expi293™ Expression Medium and transfection was achieved by mixing 2 μg of cDNA with 6 μg of polyethylenimine (PEI) and adding the mixture to two

million cells in 0.8 ml of medium. Conditioned media was harvested 4–6 days after transfection and used within 2 days. Fc constructs were characterized by SDS–PAGE to ensure protein integrity and confirm the MW of the sample. Fc-fusion proteins were quantified by WB using a known quantity of Fc-only protein. AP-fusion proteins are quantified using calf intestinal alkaline phosphatase (CIP, New England Biolabs) activity. Briefly, the activity of 20 µl of conditioned medium is compared with the activity of 1 µl of CIP at room temperature (RT) after 5 min in 50 µl of reaction in Expi293 medium. All of the AP-fusion proteins expressed at levels >1 U/µl of CIP (where 1 U = 10 pg of purified CIP).

**ELISA protocol**. An ELISA-based assay was used to test the binding between ectodomain-Fc and ectodomain-AP fusions. Each data point was reproduced in three independent experiments, and each ectodomain was tested in both orientations. Fifteen µl of a solution at 3 µg/ml of mouse anti-AP (IgGAb-1 clone 8B6.18 Thermo Fisher Scientific; Waltham, MA) in 1× PBS was added to each well of 384-well plates using an automated multichannel pipette (Viaflo Assist, Integra), sealed and incubated overnight at 4 °C. The following day, plates were washed, and 5% dry milk was added as a blocking agent, which was removed after 1 h at RT using an automated microplate washer (HydroSpeed, Tecan). Subsequently, to each well, 9 µl of ecto-AP conditioned medium containing 2 µl of monoclonal mouse anti-human IgG1-HRP (2 µg/ml; Serotec; Raleigh, NC) was added using an automated plate copier (Viaflo96, Integra) along with 7 µl of ecto-Fc culture medium. Plates were sealed and incubated for 4 h at RT in the dark. Plates were subsequently washed, and 15 µl 1-Step Ultra TMB-ELISA HRP substrate was added using an automated multichannel pipette (Viaflo Assist, Integra); after 1 h incubation at RT, the absorbance at 650 nm was recorded with a Perkin Elmer EnSpire 2300 multimode plate reader. Finally, plates were scanned to obtain matching images of the 650 nm reading. Known interactors such as NRXNs/NLGNs and IgLONs/IgLONs, were among the tested pairs and worked well as positive controls. Negative control bait (AP-only protein) was used to detect false-positive interactions. Background values necessary to quantify the positive reactions were obtained by averaging at least 30 contiguous "blank" wells (within a typical plate, the mean background Abs650 for 100 negative wells was 0.07 ± 0.027). Dynamic range of a typical plate was ~50 FOB.

**Cell-surface binding assay**. HEK293T cells were grown in DMEM (Invitrogen) supplemented with 10% FBS (Invitrogen), penicillin/streptomycin, and transfected with EGFP, myc-LRRTM2, IgSF8-EGFP, and Tenascin-R-EGFP plasmids using Fugene6 (Promega). Twenty-four hours after transfection, the cells were incubated with Fc, NRXN1β-Fc, IgSF8-Fc, or Tenascin-R-Fc proteins (10 µg/ml) in DMEM supplemented with 20 mM Hepes (pH 7.4) for 1 h at RT. Following two brief washes in DMEM/20 mM Hepes pH 7.4, cells were fixed and immunostained using mouse anti-GFP (1:500; Santa Cruz), mouse anti-c-myc (1:1000; Santa Cruz), and the Cy3-conjugated anti-Fc antibody (1:1000; Jackson ImmunoResearch). Fluorophore-conjugated secondary antibodies were from Jackson ImmunoResearch or Invitrogen (used at 1:300 or 1:1000, respectively). The cells were imaged with a Leica SP8 confocal microscope (Leica Microsystems) using a 63× objective.

**Fc-protein purification**. IgSF8-Fc and Tenascin-R-Fc proteins were produced by transient transfection of HEK293T cells using PEI (Polysciences). Six hours after transfection, media was changed to OptiMEM (Invitrogen) and harvested 5 days later. Conditioned media was centrifuged, sterile-filtered, and run over a fast-flow Protein-G agarose (Thermo-Fisher) column. After extensive washing with wash buffer (50 mM Hepes pH 7.4, 300 mM NaCl, and protease inhibitors), the column was eluted with Pierce elution buffer. Eluted fractions containing proteins were pooled and dialyzed with PBS using a Slide-A-Lyzer (Pierce) and concentrated using Amicon Ultra centrifugal units (Millipore). The integrity and purity of the purified ecto-Fc proteins was confirmed with SDS–PAGE and Coomassie staining, and concentration was determined using Bradford protein assay.

**Affinity chromatography**. Affinity chromatography experiments were performed as described[60]. P1 crude MF synaptosomes were prepared as described above from mouse brains, without the myelin-removal step. For the preparation of P2 crude synaptosome extracts, ten P21–22 rat brains were homogenized in homogenization buffer (4 mM Hepes pH 7.4, 0.32 M sucrose, and protease inhibitors) using a Dounce homogenizer. Homogenate was spun at $1000 \times g$ for 10 min at 4 °C. Supernatant was spun at $14,000 \times g$ for 20 min at 4 °C. P1 containing crude MF synaptosomes and P2 crude synaptosomes were resuspended in extraction buffer (50 mm Hepes pH 7.4, 0.1 M NaCl, 2 mM CaCl₂, 2.5 mM MgCl₂, and protease inhibitors), extracted with 1% Triton X-100 for 2 h, and centrifuged at $100,000 \times g$ for 1 h at 4 °C to pellet insoluble material. Fast-flow Protein-A sepharose beads (GE Healthcare; 250 µl slurry) prebound in extraction buffer to 100 µg human Fc or IgSF8-ecto-Fc were added to the supernatant and rotated overnight at 4 °C.

Beads were packed into Poly-prep chromatography columns (BioRad) and washed with 50 ml of high-salt wash buffer (50 mM HEPES pH 7.4, 300 mM NaCl, 0.1 mM CaCl₂, 5% glycerol, and protease inhibitors), followed by a wash with 10 ml low-salt wash buffer (50 mM HEPES pH 7.4, 150 mM NaCl, 0.1 mM CaCl₂, 5% glycerol, and protease inhibitors). Bound proteins were eluted from the beads by incubation with Pierce elution buffer and TCA-precipitated overnight. The

precipitate was resuspended in 8 M urea with ProteaseMax (Promega) per the manufacturer's instruction. The samples were subsequently reduced by 20-min incubation with 5 mM TCEP0 (tris(2carboxyethyl)phosphine) at RT and alkylated in the dark by treatment with 10 mM Iodoacetamide for 20 additional minutes. The proteins were digested overnight at 37 °C with Sequencing Grade Modified Trypsin (Promega) and the reaction was stopped by acidification. MS analysis was performed as before[60].

**Mass spectrometry**. Sorted MF synaptosomes and P2 synaptosomes were collected and protein extracted, precipitated and loaded in SDS–PAGE gels. Gels were further stained with InstantBlue Protein Stain (expedeon) according to instructions. Protein lanes were cut in four sections using clean material under the laminar flow. Each section was further cut in ~1 mm³ cubes, transferred to individual microcentrifuge tubes, and incubated three times in 30% ethanol at 60 °C for 20 min to wash excess InstantBlue Protein Stain. To prepare samples for MS, each gel section was brought to RT, and submerged in 100 µl buffer containing ammonium bicarbonate at 50 mM and TCEP at 10 mM for cystine reduction. The gel samples were incubated in this solution at 37 °C for 1 h while vortexing. The liquid was removed, and replaced with 100 µl of a new solution of 50 mM ammonium bicarbonate and 50 mM iodoacetamide for cystine alkylation. The gel samples were then incubated in the solution at RT in the dark for 45 min. Then, the liquid was removed and replaced with 100 µl of 50 mM ammonium bicarbonate and 50 mM TCEP to quench alkylation reaction at RT for 30 min. The liquid was then removed, and the gel sections were washed twice with a solution of 50 mM ammonium bicarbonate. The gel sections were then each submerged by a 100 µl solution of 50 mM ammonium bicarbonate and 1 µg of trypsin, and incubated overnight at 37 °C with vortexing for protein digestion to peptides. The next day, the liquid from the digested gel samples was collected in new microcentrifuge tubes. Then the gel slices were treated a total of three times with a solution of 50% acetonitrile, 5% formic acid for 20 min followed by collection of the liquid in the new tube. The tubes now containing the extracted peptides from the gel slices were dried by vacuum centrifugation. The dried peptides were then resuspended in 200 µl of a solution of 0.5% trifluoroacetic acid and desalted on C18 resin spin columns (Pierce), using 5% acetonitrile, 0.5% trifluoroacetic acid to wash, and 80% acetonitrile, 0.5% formic acid buffer to elute. These desalted samples were then dried by vacuum centrifugation and resuspended in LC–MS/MS sample buffer of 5% acetonitrile, 1% formic acid.

For MS data acquisition from the samples, 3 µg of peptides corresponding to each of the four gel sections of each of the two samples from each experiment was injected into a Thermo Orbitrap Fusion Mass Spectrometer equipped, with an Ultimate 3000 RSLCnano LC. Peptides were loaded onto a vented Acclaim Pepmap 100, 75 µm × 2 cm, nanoViper trap column, and subsequently separated using a nanoViper analytical column (Thermo-164570, 3 µm, 100 Å, C18, 0.075 mm, 500 mm) and electrosprayed with stainless steel emitter tip assembled on the Nanospray Flex Ion Source with a spray voltage of 2000 V. Buffer A contained H₂O with 5% ACN and 0.125% FA, and buffer B contained H2O with 95% ACN and 0.125% FA. The chromatographic run was for 160 min in total with the following profile of percent buffer B: 0 to 8% for 6 min, to 24% for 64 min, to 36% for 26 min, to 55% for 10 min, to 95% for 10 min, held at 95% for 20 min, then brought down to 2% over 1 min, and held at 2% for 29 min. Additional MS parameters include: ion transfer tube temp at 300 °C, positive spray voltage at 2500 V, and top speed with a cycle time of 3 s. MS1 scan in Orbitrap with 120 K resolution, scan range = 300–1800 (m/z), max injection time = 50 ms, AGC target = 100,000, microscans = 1, RF lens = 60%, and datatype = centroid. MIPS mode = peptide, included charge states = 2–8 (reject unassigned). Dynamic exclusion enabled with n = 1 for 30 s exclusion duration at 5 p.p.m. for high and low. Precursor selection decision for MS2 scan in the ion trap set at most intense with a 1000 minimum intensity threshold, isolation window = 1.6, scan range = auto normal, first mass = 100, collision energy 32% HCD, IT scan rate = rapid, max injection time at 50 ms, AGC target = 10,000, datatype = centroid, inject ions for all available parallelizable time.

Spectrum raw files from samples were extracted into ms1 and ms2 files using in-house program RawConverter (http://fields.scripps.edu/downloads.php)[116], and the tandem mass spectra were searched against UniProt mouse protein database (downloaded on 03-25-2014; The UniProt Consortium 2015) and matched to sequences using the ProLuCID/SEQUEST algorithm (ProLuCID ver. 3.1)[117,118] with 50 p.p.m. peptide mass tolerance for precursor ions and 600 p.p.m. for fragment ions. The search space included all fully and half-tryptic peptide candidates that fell within the mass tolerance window with no miscleavage constraint, assembled and filtered with DTASelect2 (ver. 2.1.3)[119,120] through Integrated Proteomics Pipeline (IP2 v.3, Integrated Proteomics Applications, Inc., CA, USA http://www.integratedproteomics.com). To estimate peptide probabilities and false discovery rates (FDR) accurately, we used a target/decoy database containing the reversed sequences of all the proteins appended to the target database. Each protein identified was required to have a minimum of one peptide of minimal length of five amino acid residues and within 10 p.p.m. of the expected m/z. However, this peptide had to be an excellent match with an FDR < 0.001 and at least one excellent peptide match. After the peptide/spectrum matches were filtered, we estimated that the protein FDRs were ≤1% for each sample analysis.

To identify the unique protein composition of MF synapses, first we filtered our protein list on the criteria that a candidate protein had to have at least three peptide identifications among the six samples. Then Log2 fold change of sorted MF synaptosomes vs the P2 synaptosome background was calculated on the basis of the measured NSAF. This measure normalizes spectral counts for a protein based on the total number of spectra in the run, the length of the protein, and the total number of peptides that comprise it[121,122]. Proteins that were not identified in a sample after filtering had their NSAF values imputed as 0 for all calculations. Proteins were determined significant if the Student's $t$ test $p$ value was <0.05 among the three replicates. A $q$ value was calculated based on Benjamini–Hochberg procedure and the highest ranked protein with a $q$ value < 0.05 was used as a cutoff for determining high-confidence measurements at a 5% FDR.

**Gene ontology analysis**. To find cellular component terms overrepresented in sorted MF synaptosomes or P2 synaptosomes, we used the statistical over-representation test in the Panther Classification System using the mouse genome as reference (http://www.pantherdb.org/)[123]. To analyze the distribution of the MF synaptic proteome within major cellular components, we used the functional classification tool in Panther. To select CSPs among the MF synaptic proteome, we manually queried UniProt (https://www.uniprot.org/) for annotated proteins with transmembrane and extracellular domains, as well as secreted and ECM proteins. CSP functional class and protein domains were also manually examined using UniProt and known literature in PubMed (https://www.ncbi.nlm.nih.gov/pubmed/).

**Fc pull-down assays**. For pull-down assays on HEK293T cells, cells were grown in 10 cm dishes in DMEM (Invitrogen) supplemented with 10% FBS (Invitrogen) and penicilin/streptomycin, and transfected with EGFP, IgSF8-EGFP, or Tenascin-R-EGFP expression constructs using Fugene6 (Promega). Twenty-four hours after transfection, the media was changed to OptiMEM (Invitrogen) for 2 h. Cells were then lysed in 1 ml ice-cold RIPA buffer (20 mM Tris HCl pH 7.5, 150 mM NaCl, 5 mM EDTA, 1% Triton X-100, and protease inhibitors (Roche)) for 1 h at 4 °C on a rocking platform. Lysates were spun at $17,000 \times g$ for 30 min at 4 °C. Three μg of human Fc, IgSF8-Fc, or Tenascin-R-Fc was added to 1 ml of supernatant and rotated overnight at 4 °C. Protein-A agarose beads (50 μl slurry) were added and rotated for 1 h at 4 °C. Beads were washed three times in cold RIPA buffer and once in PBS, boiled in 50 μl 2× sample buffer, and analyzed by western blotting using a mouse anti-GFP primary antibody (1:1000; Santa Cruz) and HRP-conjugated goat anti-mouse IgG secondary antibody (1:10,000; Thermo-Fisher). Uncropped and unprocessed scans are provided in Source data file.

**Subcellular fractionation**. Synaptic fractionation was based on a previously described method[124]. Ten P21 rat brains were homogenized in 12 ml per brain with homogenization buffer (0.32 M sucrose, 4 mM Hepes pH 7.4, 1 mM MgCl$_2$, and protease inhibitors; homogenate), centrifuged at $1500 \times g$ for 15 min, and the supernatant was collected (postnuclear supernatant). The supernatant was then centrifuged at $18,000 \times g$ for 20 min, and the resulting supernatant (cytosol) and pellet (crude membrane) collected. The pellet was resuspended in homogenization buffer and loaded onto 0.85/1.0/1.2 M discontinuous sucrose gradients, and centrifuged at $78,000 \times g$ for 120 min. The material at the 1.0/1.2 M interface was collected (synaptosome). Triton X-100 was added to 0.5% and extracted at 4 °C by end-over-end agitation for 20 min. The extract was centrifuged at $32,000 \times g$ for 20 min, the supernatant collected (soluble synaptosome/Triton-soluble fraction) and the pellet was resuspended in homogenization buffer, loaded onto a 1.0/1.5/2.0 M sucrose gradient, and centrifuged at $170,000 \times g$ for 2 h. Material was collected at the 1.5/2.0 M interface (PSD). Triton X-100 (0.5%) was added and detergent-soluble material extracted at 4 °C by end-over-end agitation for 10 min. Lastly, the extract was centrifuged at $100,000 \times g$ for 20 min and the pellet resuspended in homogenization buffer (purified PSD/Triton-insoluble fraction). Primary antibodies used were the following: mouse anti-PSD95 (1:2500; Thermo-Fisher), rabbit ant-Synaptophysin (1:2500; Sigma-Aldrich) and goat anti-IgSF8 (1:500; R&D Systems). HRP-conjugated secondary antibodies were from Thermo-Fisher (1:10000). Uncropped and unprocessed scans are provided in Source data file.

**Immunocytochemistry**. Crude MF (PI fraction) and small hippocampal (SI fraction) synaptosomes were spun at $15,000 \times g$ for 20 min at 4 °C after being collected, resuspended in PBS, transferred to PDL-coated eight-well Nunc Lab-Tek chamberslides (Thermo-Fisher) and incubated at 4 °C on a shaker for 90 min to allow synaptosomes to settle. After, synaptosomes were fixed in 2% paraformaldehyde (PFA) and stained. Briefly, synaptosomes were blocked and permeabilized in 3% BSA, 0.1% Saponin in PBS (BSA-BLOCK) at RT for 30 min and incubated with primary antibodies in BSA-BLOCK at 4 °C overnight. After, synaptosomes were washed three times with PBS, incubated with secondary antibodies at RT for 1 h, washed three times with PBS, incubated with Hoechst dye at RT for 5 min, washed again, and mounted with Mowiol-4-88 (MilliporeSigma Calbiochem). For live labeling, crude MF synaptosomes were incubated in non-permeabilizing conditions with primary antibodies diluted in 3% BSA in PBS before blocking, permeabilization, and staining of intracellular epitopes. In sorting experiments, a sample of myelin-depleted presorting P1 MF synaptosomes was

collected before sorting. Remaining sample was sorted and spun at $1000 \times g$ for 10 min at 4 °C to pellet-sorted MF synaptosomes. Finally, sorted MF synaptosomes were resuspended in PBS and stained in permeabilizing conditions as above, in parallel to myelin-depleted presorting MF synaptosomes. Primary antibodies were the following: rat anti-Nectin3 (1:100; Hycult Biotech), mouse anti-GluK5/KA2 (1:200; NeuroMab), rabbit anti-Synpr (1:5000; Synaptic Systems), guinea pig anti-VGluT1 (1:5000; Millipore), and mouse anti-PSD95 (1:250; Thermo-Fisher). Fluorophore-conjugated secondary antibodies were from Jackson ImmunoResearch or Invitrogen (used at 1:300 or 1:1000, respectively). Confocal images were acquired using a Leica SP8 confocal microscope (Leica Microsystems) and analyzed with Fiji[125].

**Immunohistochemistry**. P28 WT mice were anesthetized by intraperitoneal injection with a lethal dose of 1 μl/g xylazine (VMB Xyl-M 2%), 2 μl/g ketamine (Eurovet Nimatek, 100 mg/ml), and 3 μl/g 0.9% saline. Next, mice were transcardially perfused with 4% PFA, in PBS. Brains were dissected, postfixed in 4% PFA in PBS at 4 °C for 1 h, and cryopreserved in 30% sucrose overnight at 4 °C. After, perfused brains were embedded in Tissue-Tek O.C.T. compound (Sakura) and frozen in isopentane at −55 to −65 °C. For fresh frozen immunohistochemistry, mice were euthanized using isoflurane (Halocarbon). Next, brains were quickly dissected, embedded in Tissue-Tek O.C.T. compound (Sakura), and frozen in isopentane at −55 to −65 °C. To prepare hippocampal sections, we used the cryostats (NX70, Thermo Fisher) and CM3050 S (Leica Biosystems). Frozen brains were cut and 16–20 μm thick coronal sections were collected on SuperFrost Ultra Plus adhesion slides (Thermo-Fisher). The fresh frozen sections were postfixed with 1:1 acetone–methanol for 5 min at −20 °C and quickly washed in PBS before staining.

In some cases, heat-induced antigen retrieval (H-AR) was done before immunostainings in sections of PFA perfused brains. For H-AR, sections were submerged in sodium citrate buffer containing 0.05% Tween20 and 10 mM sodium citrate pH of 6.0, and heated in either a microwave up to boiling, for 30 s, twice for 15 s and once for 10 s with a 1–2 min cooling period at RT between heating; or in the 2100 Antigen Retriever pressure cooker (Aptum Biologics Ltd) for 25 min up to 120 °C. After H-AR, sections were washed in PBS. Both fresh frozen and perfused sections were then permeabilized at RT for 20 min in PBS containing 0.05% Triton X-100 (Sigma). Sections were then blocked at RT for 2 h in PBS containing 10% normal horse serum (NHS), 0.5% Triton X-100, 10% glycine 2 M, 0.02% gelatin, and, in the case of using mouse primary antibodies, 1:50 donkey anti-mouse IgG antigen-binding fragments (Jackson ImmunoResearch). Sections were then washed in PBS-0.5% Triton X-100 at RT and incubated at 4 °C overnight with primary antibodies in PBS containing 5% NHS, 0.5% Triton X-100, and 0.02% gelatin. Afterward, sections were washed in PBS-0.5% Triton X-100 at RT before a 2 h incubation at RT, with secondary antibodies in PBS containing 5% NHS, 0.5% Triton X-100, and 0.02% gelatin. Before mounting coverslips with Mowiol-4-88 (MilliporeSigma Calbiochem), sections were washed in PBS-0.5% Triton X-100. Primary antibodies were the following: rat anti-Nectin3 (1:100; Hycult Biotech), mouse anti-GluK5/KA2 (1:500; NeuroMab), rabbit anti-Synpr (1:5000; Synaptic Systems), guinea pig anti-VGluT1 (1:5000; Millipore), rabbit anti-Synapsin3 (1:500; Synaptic Systems), rabbit anti-Afadin (1:500; Thermo-Fisher), chicken anti-ZnT3 (1:500; Synaptic Systems), rabbit anti-mGluR2/3 (1:500; Merck Millipore), rabbit anti-CRTAC1 (1:50; Merck Millipore), goat anti-IgSF8 (1:500; R&D Systems), mouse anti-PSA-NCAM (1:200; Merck Millipore), rat anti-NCAM2 (1:200; R&D Systems), sheep anti-ISLR2 (1:200; R&D Systems), goat anti-NEGR1/Kilon (1:500; R&D Systems), mouse anti-NPTX1 (1:200; BD Biosciences), goat anti-NRP1 (1:200; R&D Systems), goat anti-NRP2 (1:200; R&D Systems), sheep anti-Noelin (1:500; R&D Systems), goat anti-PlexinA3 (1:50; Thermo-Fisher), rabbit anti-ROBO2 (1:200; Aviva Systems), goat anti-Neogenin (1:200; R&D Systems), sheep anti-SALM2 (1:200; R&D Systems), rabbit anti-FAM171A2 (1:50; ThermoFisher), mouse anti-Tenascin-R (1:200; R&D Systems), sheep anti-Contactin 1 (1:200; R&D Systems), goat anti-CD200 (1:100; R&D Systems), goat anti-ICAM5 (1:200; Novus), goat anti-Kit/SCFR (1:100; R&D Systems), mouse anti-LSAMP (1:1000; Developmental Studies Hybridoma Bank), mouse anti-Trk-C (1:200; R&D Systems), goat anti-PlexinA1 (1:200; R&D Systems), and goat anti-PTPRS (1:200; R&D Systems). Fluorophore-conjugated secondary antibodies were from Jackson ImmunoResearch or Invitrogen (used at 1:300 or 1:1000, respectively).

For free-floating immunohistochemistry, mouse anesthesia, perfusion, and brain dissection were performed as described above. Postfixation was done overnight at 4 °C in PBS containing 4% PFA. After, brains were washed three times with PBS at RT and embedded in 3% agarose. After, 80 μm thick sections were collected in PBS containing 0.2% NaN3 using the Vibrating Microtome 7000 (Campden Instruments LTD). Vibratome sections were washed at RT in PBS-0.5% Triton X-100. Next, sections were blocked at 4 °C overnight in PBS containing 10% NHS, 0.5% Triton X-100, 0.5 M glycine, and 0.2% gelatin. Thereafter, sections were incubated overweekend with primary antibodies in PBS containing 5% NHS, 0.5% Triton X-100, and 0.2% gelatin. Sections were then washed with PBS-0.5% Triton X-100 and incubated with the secondary antibodies in PBS containing 5% NHS, 0.5% Triton X-100, and 0.2% gelatin, at 4 °C overnight. Subsequently, sections were washed in PBS-0.5% Triton X-100. To stain nuclei, sections were incubated in PBS containing Hoechst (1:200) for 10 min at RT. Finally, sections were washed in PBS, collected on a microscope slide, and coverslips were mounted with Mowiol-4-88

(MilliporeSigma Calbiochem). Primary antibodies were the following: chicken anti-GFP (1:500; Aves Labs), rabbit anti-Synpr (1:2500; Synaptic Systems), and goat anti-IgSF8 (1:500; R&D Systems). Fluorophore-conjugated secondary antibodies were from Jackson ImmunoResearch or Invitrogen (used at 1:300 or 1:1000, respectively). Images were acquired using a Leica SP8 confocal microscope (Leica Microsystems) or Slide Scanner Axio Scan.Z1 (Zeiss) and analyzed with Fiji[125].

**Western blots.** Sorted MF synaptosomes were collected into sorting tubes containing 4× lysis buffer (40 mM Hepes, 600 mM NaCl, 4% NP-40, 4% sodium deoxycholate, 0.4% SDS, and 20 mM EDTA). P2 synaptosomes were resuspended in 2× lysis buffer, and both samples incubated on ice for 30 min and further rotated at 4 °C for 1 h. To precipitate proteins, 20% (v/v) of freshly prepared TCA was added to the lysate of both samples. After, lysates were vortexed and incubated on ice overnight in the cold room. Finally, precipitates were spun at 13,000 × g at 4 °C for 30 min, washed three times with ice-cold acetone and air-dried. Protein precipitates were then resuspended in 4× Laemmli buffer (8% SDS, 40% glycerol, 20% β-mercaptoethanol, 0.01% bromophenol blue, and 250 mM Tris HCl pH 6.8), pH-adjusted with 1.5 M Tris HCl pH 8.8, spun, boiled at 95 °C for 5 min, and loaded in a gel for SDS–PAGE. Primary antibodies were the following: mouse anti-PSD95 (1:2500; Thermo-Fisher), rabbit ant-Synaptophysin (1:2500; Sigma-Aldrich), rabbit anti-Synpr (1:2500; Synaptic Systems), rabbit anti-Nectin3 (1:500; Abcam), goat anti-MBP (1:500; Santa Cruz), rabbit anti-Histone H3 (1:1000; Cell Signaling Technology), goat anti-IgSF8 (1:2000; R&D Systems), mouse anti-TenR (1:1000; R&D Systems), rabbit anti-BRINP2 (1:100; Atlas Antibodies), sheep anti-CNTN1 (1:500; R&D Systems), goat anti-ICAM5 (1:500; R&D Systems), sheep anti-ISLR2 (1:1000; R&D Systems), goat anti-Kit/SCFR (1:1000; R&D Systems), goat anti-NEGR1 (1:1000; R&D Systems), goat anti-Neogenin (1:1000; R&D Systems), goat anti-NRP1 (1:500; R&D Systems), goat anti-PlexinA1 (1:500; R&D Systems), rabbit anti-PTPRD (1:500; Novus), goat anti-PTPRS (1:1000; R&D Systems), rabbit anti-ROBO2 (1:1000; Abcam), and sheep anti-Teneurin-4 (1:1000; R&D Systems). HRP-conjugated secondary antibodies were from Thermo-Fisher (1:10,000). Uncropped and unprocessed scans are provided in Source data file.

**Direct binding assay.** For direct binding assays of IgSF8 to Tenascin-R and Brevican, 1 μg recombinant His-tagged mouse IgSF8 (Sino Biological) was incubated in 1 ml binding buffer (10 mM HEPES pH 7.4, 150 mM NaCl, 2 mM CaCl₂, 1 mM MgCl₂, and 0.1% Tween-20) with equimolar amounts of control Fc-protein (Jackson ImmunoResearch), purified Tenascin-R-Fc or Brevican-Fc, and rotated end-over-end for 1 h at RT. Protein-A/G agarose beads (100 μl slurry; Santa Cruz Biotechnology) were added and rotated end-over-end for 1 h at 4 °C. Beads were washed 4× in binding buffer and 1× in PBS, and boiled in 50 μl 2× sample buffer. Samples were analyzed by WB. Rabbit anti-6x-His Tag primary antibody (1:250; Thermo-Fisher) and HRP-conjugated anti-rabbit secondary antibody (1:10,000; Thermo-Fisher) were used to detect IgSF8-His. Uncropped and unprocessed scans are provided in Source data file.

**Biolayer Interferometry analysis.** BLI binding experiments were conducted using a BLItz instrument (ForteBio, Menlo Park, CA) at RT. Anti-human Fc capture Biosensors were pre-wetted for 10 min in 300 ml of 10 mM Hepes, pH 7.4, 150 mM NaCl, 1 mM CaCl2, 0.2% Tween20, and 1% (w/v) BSA prior to use. Subsequently, the sensor tips were incubated for 10 min with conditioned medium of EXPI293 cells transfected with TenR-Fc. The binding reaction occurred under agitation in a 4 ml drop containing various concentrations of purified proteins. Both association and dissociation were allowed to occur for 120 s. Technical triplicates were performed for each concentration and averages were used for the final calculations. Nonspecific binding and instrument noise were subtracted by using a sensor tip saturated with Fc fragment alone. A slight systemic drifting is apparent in the curves, but it did not impact the quality of the experiment as judged by the statistical quality of the plot in C. The pre-wetting buffer was used for the association step, while dissociation was performed in 20 mM HEPES pH 8, 150 mM NaCl, 3 mM EDTA, 0.2% Tween20, and 0.1% BSA.

**In utero electroporation.** For in vivo IgSF8 localization studies, hippocampi of E15.5-day-old embryos of timed-pregnant WT C57BL/6 J mice were unilaterally electroporated with FUGW-mGFP-T2A-HA-IgSF8 plasmid. Briefly, the dam was anesthetized with isoflurane and the uterus exposed. A solution of 1 μg/μl DNA and 0.01% fast green dye was injected into the embryonic lateral ventricle with a bevelled glass capillary. The embryo's head was then placed between the paddles of pair of platinum tweezer-type electrodes (Napagene) with the cathode lateral to the filled ventricle and seven 50 ms, 40 V pulses were delivered at 1 Hz by a ECM830 electroporator (Harvard Apparatus). After electroporation, the uterus was replaced, the incision sutured closed and the dam allowed to give birth normally.

**Image acquisition and analysis of in utero electroporated mice.** In utero electroporated mice were transcardially perfused with ice-cold 4% PFA at P28. Brains were dissected and postfixed for 1 h at 4 °C. Brains were then embedded in 3% agarose (Sigma-Aldrich) and 60 μm coronal sections were cut on a vibratome (VTS1000S, Leica Biosystems). Sections in which sparsely labeled MF axons visibly expressed GFP were immunostained with chicken anti-GFP (1:500; Aves Labs) and

rat anti-HA (1:250; Roche) primary antibodies. Fluorophore-conjugated secondary antibodies were from Jackson ImmunoResearch or Invitrogen (used at 1:300 or 1:1000, respectively). Coverslips were then mounted using Prolong Gold Antifade (ThermoScientific).

Sparsely labeled MF boutons were imaged on a Leica SP8 confocal microscope (Leica Microsystems) using a 63× oil immersion objective and 1.75 zoom factor. Z-stack containing the complete MF bouton structure, including filopodia were acquired using a z-step of 1 μm. Images were analyzed with Fiji.

**Transmission electron microscopy.** To analyze crude MF synaptosomes, the P1 fraction collected from ten mice was fixed with 2% glutaraldehyde, 4% PFA, 0.2% picric acid in 0.1 M PB pH 7.40, gelatin embedded, and 100 μm vibratome sections were incubated in 1% osmium tetroxide, 1.5% potassium hexacyanoferrate in 0.1 M cacodylate buffer, and then dehydrated with ethanol. Subsequently, sections were contrasted in 2% uranyl acetate, and en block lead acetate, washed, and embedded in Epon. Samples were sectioned as 70 nm, collected on copper grids, and examined with a Jeol Jem 1400 electron microscope.

To analyze morphological changes in *Rbp4-Cre:Igsf8* cKO double transgenic mice, P30 littermates (*Rbp4-Cre⁺:Igsf8⁺/⁺ and Rbp4-Cre⁺:Igsf8^f/f*) were transcardially perfused with 4% PFA, 2.5% glutaraldehyde, and 0.2% picric acid in 0.1 M PB. Brains were removed, embedded in 3% low gelling temperature agarose (Sigma-Aldrich) and 80 μm sections were cut on a vibratome (VTS1000S, Leica Biosystems). Selected sections were then postfixed in a solution of 1% OsO₄ containing 1.5% potassium ferrocyanide at RT for 60 min and incubated overnight with 0.5% uranyl acetate and 25% methanol at 4 °C. Then the sections were stained with Walton's lead aspartate for 30 min at 60 °C, dehydrated by subsequent ethanol series with increasing concentration, and finally infiltrated and embedded in resin (epon 812). Ultrathin 70 nm sections were cut using an ultramicrotome (Leica Biosystems EM UC7) and collected on copper grids. Sections were imaged using a JEOL JEM1400 TEM equipped with an Olympus SIS Quemesa camera operated at 80 kV. MF-CA3 synapses were identified by their morphology, presence of multiple PSDs, and high synaptic vesicle content. Overview images were taken at 2500× to analyze bouton number and area, and 5000× to analyze ultrastructure. Images were analyzed in MIB2 software (University of Helsinki) to segment the area and perimeter of individual MF-CA3 synapses, and the number and length of PSDs and AZs. Segmented images were analyzed with ImageJ software using a custom-made script. Analysis were conducted blind to genotype.

**Lentivirus production.** For lentivirus production, HEK293T cells were transfected with mCherry or Cre-T2A-mCherry containing FUGW vector plasmids, and helper plasmids PAX2 and VSVG using Fugene6 (Promega). Supernatant was collected 48 h after transfection, spun at 1000 × g to remove debris and filtered through a 0.22 μm filter (Millipore). A total of 200 μl aliquots were stored at −80 °C.

**Adeno-associated virus production and purification.** HEK293T cells were seeded in DMEM (Invitrogen) containing 10% FBS (Invitrogen). Transfection mix, containing PEI and OptiMEM (Invitrogen), was incubated with OptiMEM containing pΔF6 helper, pAAV V2/9 Rep/Cap and pAAV-hsyn1-mGFP-T2A-Cre, pAAV-hsyn1-mGFP or pAAV-Ef1a-DIO-hChR2(E123T/T159C)-EYFP plasmids for 20 min at RT. Prior to adding DNA:PEI mix to HEK293T cells, DMEM-10% FBS was carefully replaced with DMEM-1% FBS. After 5 h, DMEM-10% FBS was added to each plate. Three days after transfection, cells were harvested, pooled, centrifuged at 1000 × g at 4 °C for 10 min and cell pellets lysed in lysis buffer (150 mM NaCl and 50 mM Tris HCl-pH 8.5 in endotoxin free H₂O). Lysates were frozen in dry ice and ethanol, and thawed at 37 °C in a water bath, for three times. Thereafter, supernatants were collected and Benzonase (Sigma) was added to a final concentration of 50 U/ml. After an incubation of 30 min at 37 °C, lysates were centrifuged at 5000 × g for 20 min at RT. Supernatant was then collected through a 0.45 μm filter (Thermo-Fisher) and carefully layered onto iodixanol gradients in 25 × 77 mm OptiSeal tubes (Beckman Coulter). Gradients were prepared using OptiPrep iodixanol (Sigma), 5 M NaCl, 5× PBS with 1 mM MgCl₂ and 2.5 mM KCl (5× PBS-MK), and sterile H₂O. Gradients were centrifuged at 300,000 × g and 12 °C for 100 min in the Optima XE-100 Ultracentrifuge (Beckman Coulter). Next, AAVs were collected with an 18 G needle (Beckman Coulter) from between the 40 and 60% layers, and diluted in 5 ml 1× PBS-MK. The diluted AAVs were then desalted and concentrated by centrifugation at 5000 × g for 30 min at 20 °C in a prerinsed Amicon Ultra-15 filter (Millipore) in 1× PBS-MK. Finally, concentrated and desalted AAVs were washed by centrifugation at 5000 × g and 20 °C for 5 min with PBS containing 0.01% Pluronic F68 (Thermo-Fisher), aliquoted and stored at −80 °C.

**Stereotactic injections.** For 3D reconstruction of MF boutons, *Igsf8* cKO mouse littermates were injected with AAV vectors expressing mGFP or mGFP-T2A-Cre in the DG at P7. For electrophysiology experiments, *Rbp4-Cre;Igsf8* cKO mouse littermates were injected with AAV vectors expressing Cre-dependent hChR2 (E123T/T159C)-EYFP in the DG at P7. Prior to stereotactic injections, mice were intraperitoneally injected with 0.05 mg/kg buprenorphine (Vetergesic). After 1 h, mice were anesthetized with 5% isoflurane and Duratears was applied to the eyes to

prevent them from drying out. Mice were placed in a mouse stereotact (KOPF) equipped with a neonatal mouse adaptor (Stoelting). During the rest of the procedure 2.5% isoflurane was constantly administered. After shaving and disinfecting the mouse's head, local anesthesia was administered by a subcutaneous injection with 100 µl lidocaine (xylocain 1%). After 5 min an incision was made in the skin. Thereafter, the injector (Nanoject III Drummond) was placed at predetermined coordinates and beveled capillaries that penetrate skull and skin were used. One minute after lowering the capillary into the brain, viral mix was injected in the DG at 1 nl/s. After a 5 min recovery, the capillary was pulled out at ~0.1 mm/5 s. The incision was stitched with surgical glue (Millpledge Veterinary). After 6 h, their health was examined and mice were injected with 0.1 mg/kg buprenorphine.

**Image acquisition and analysis of injected mice.** Injected mice were perfused at P28 and brains dissected, postfixed, embedded, and sectioned as before. Sections in which sparsely labeled MF axons visibly expressed GFP were immunostained with a chicken anti-GFP primary antibody (1:500; Aves Labs) and fluorophore-conjugated anti-chicken secondary antibody (1:1000; Invitrogen). Coverslips were then mounted using Mowiol-4-88 (MilliporeSigma Calbiochem). Sparsely labeled MF boutons were imaged on a Leica SP8 confocal microscope (Leica Microsystems) using a 63× oil immersion objective and 1.75 zoom factor. Z-stacks containing the complete MF bouton structure and filopodia were acquired using a z-step of 0.2 µm. Images were acquired from the CA3 hippocampal subregion. Acquired Z-stacks were then individually opened in Fiji[125] and registered to correct possible drift in z-axis using the StackReg plugin. In order to reconstruct individual MF boutons, aligned stacks were then uploaded in Imaris (Bitplane). The number of filopodia was manually measured in 3D and only filopodia with a minimum length of 1.5 µm were counted[89]. The 3D reconstructions of the MF bouton core were obtained by initially segmenting the volume containing the entire MF terminal. The segmented volume was subsequently masked and the MF bouton was reconstructed using the "New Surface" tool. Analysis of the volume and surface values was acquired via the "Statistics" function. Morphological analyses were conducted blind to conditions.

**Acute slice electrophysiology.** Sagittal slices were prepared from postnatal day P27–35 Rbp4-Cre;Igsf8 cKO littermates. Briefly, after decapitation, the brain was quickly removed and transferred into ice-cold cutting solution (in mM): 87 NaCl, 2.5 KCl, 1.25 $NaH_2PO_4$, 10 glucose, 25 $NaHCO_3$, 0.5 $CaCl_2$, 7 $MgCl_2$, 75 sucrose, 1 kynurenic acid, 5 ascorbic acid, 3 pyruvic acid, pH 7.4 with 5% $CO_2/95\%$ $O_2$), and whole brain sagittal slices (300 µm) were cut using a vibratome (VT1200, Leica Biosystems). Afterward, slices were transferred to 34 °C cutting solution for 45 min to recover and finally maintained at RT until used for recordings.

For recordings, brain slices were continuously perfused in a submerged chamber (Warner Instruments) at a rate of 3–4 ml/min with (in mM): 119 NaCl, 2.5 KCl, 1 $NaH_2PO_4$, 26 $NaHCO_3$, 4 $MgCl_2$, 4 M $CaCl_2$, 11 glucose at pH 7.4 with 5% $CO_2/$ 95% $O_2$. For paired-pulse and spontaneous release recordings, 20 µM bicuculline, 100 µM AP-5, and 150 nM CNQX was added to the ACSF. Whole-cell patch-clamp recordings were done using borosilicate glass recording pipettes (resistance 3.5–5 MΩ). For paired-pulse and spontaneous release experiments, we used the following internal solution (in mM): 115 CsMSF, 20 CsCl, 10 HEPES, 2.5 $MgCl_2$, 4 ATP, 0.4 GTP, 10 creatine phosphate, and 0.6 EGTA (pH 7.25). For E/I ratios, we used (in mM): 132 CsMSF, 8 CsCl, 10 HEPES, 0.5 mM $CaCl_2$, 1 EGTA, 10 glucose, and 5 QX-314 (pH 7.3). For experiments testing eEPSP-induced action potential generation, we used the following internal medium (in mM): 135 KGluconate, 4 KCl, 2 NaCl, 10 HEPES, 4 EGTA, 4 MgATP, and 0.3 NaATP (pH 7.25).

Spontaneous and evoked input to CA3 pyramidal neurons was recorded by whole-cell voltage-clamp recordings ($V_m = -70$ mV) from visually identifiable CA3 pyramidal neurons, using a Multiclamp 700B amplifier (Axon Instruments). Spontaneous input was analysed using Mini Analysis program (Synaptosoft), evoked data were analyzed using Clampfit 10.7 (Axon Instruments). Stereotactic injections of pAVV-Ef1a-DIO-ChR2-EYFP in the DG of Rbp4-Cre;Igsf8 cKO littermates at P7 resulted in expression of ChR2 only in GCs of the DG, and therefore allowed light-induced activation specifically of the MF pathway. We used site-directed, region-controlled activation (UGA-42 Geo of Rapp OptoElectronics) of MFs within SL area close to the recorded neurons. We typically needed 2–10% laser power (0.5–2.6 mW with open aperture) for 1–2 ms, using circular area selections between 40 and 120 (32–96 µm diameter) to elicit baseline eEPSCs responses (300–400 pA). Recordings requiring significantly more optical stimulation to elicit eEPSCs were discarded. For paired-pulse ratio experiments ($V_m = -70$ mV), paired optical stimulations (interstimulus interval (ISI): 25, 50, 100, 200, 400, 1000, and 2000 ms) were delivered every 20 s (each ISI was repeated four times) and calculated as the eEPSC2/eEPSC1 ratio. Inhibition/excitation ratios were recorded by separating eEPSCs or eIPSCs using optical stimulation when the CA3 neuron was clamped at the reversal potential of the inhibitory (−70 mV) or excitatory (0 mV) input, respectively. Single eEPSCs and eIPSCs were repeated eight times and averaged. For testing eEPSP-induced action potential generation, we established a basal eEPSC amplitude of ~400 pA, while keeping the neuron at the chloride reversal potential ($V_m = -82.6$ mV). Subsequently, while recorded neurons were in current clamp (Vrest), train stimulations (10 Hz, 20 stimulations) were used to elicit MF activation to monitor CA3 pyramidal cell excitability. For all measurements, we performed a minimum of three independent preparations.

**Quantification and statistical analysis.** Data analysis were performed in GraphPad Prism 8 (GraphPad software), Clampfit 10.7 (Axon Instruments), MiniAnalysis (Synaptosoft), and Fiji (NIH). For quantification, datasets were tested for normality using D'Agostino and Pearson test. If datasets passed the test, they were analyzed using Student's unpaired t test. Otherwise, the datasets were analyzed using nonparametric unpaired t tests (Mann–Whitney).

### Resource table

| Antibodies | | |
|---|---|---|
| Chicken anti-GFP | Aves Labs | Cat#: GFP-1010; RID: AB_2307313 |
| Mouse anti-GFP clone B-2 | Santa Cruz | Cat#: SC-9996; RRID: AB_627695 |
| Mouse anti-c-myc clone 9E10 | Santa Cruz | Cat#: SC-40; RRID: AB_627268 |
| Rabbit anti-6x-His Tag | Thermo-Fisher | Cat#: PA1-983B; RRID: AB_1069891 |
| Goat anti-MBP clone D-18 | Santa Cruz | Cat#: SC-13912; RRID: AB_648794 |
| Mouse anti-PSD95 clone 7E3-1B8 | Thermo-Fisher | Cat#: MA1-046; RRID: AB_2092361 |
| Rabbit anti-Histone H3 clone D1H2 | Cell Signaling Technology | Cat#: 4499; RRID: AB_10544537 |
| Rabbit anti-Synaptophysin clone SVP-38 | Sigma-Aldrich | Cat#: S5768; RRID: AB_477523 |
| Rat anti-Nectin3 clone 103-A1 | Hycult Biotech | Cat#: HM1053; RRID: AB_533278 |
| Rabbit anti-Nectin3 | Abcam | Cat#: AB63931; RRID: AB_1142394 |
| Mouse anti-GluK5/KA2 clone N279B/27 | NeuroMab | Cat#: 75-362; RRID: AB_2315855 |
| Rabbit anti-Synaptoporin | Synaptic Systems | Cat#: 102 002; RRID: AB_887841 |
| Guinea pig anti-VGluT1 | Merck Millipore | Cat#: AB5905; RRID: AB_2301751 |
| Rabbit anti-Synapsin3 | Synaptic Systems | Cat#: 106 303; RRID: AB_2619775 |
| Rabbit anti-Afadin clone A7L9H48 | Thermo-Fisher | Cat#: 700193; RRID: AB_2532299 |
| Chicken anti-ZnT3 | Synaptic Systems | Cat#: 197 006; RRID: AB_2725754 |
| Rabbit anti-mGluR2/3 | Merck Millipore | Cat#: AB1553; RRID: AB_11212089 |
| Rabbit anti-CRTAC1 | Merck Millipore | Cat#: ABD80 |
| Rabbit anti-BRINP2 | Atlas Antibodies | Cat#: HPA061920; RRID: AB_2684639 |
| Goat anti-IgSF8 | R&D Systems | Cat#: AF3117; RRID: AB_2233385 |
| Rabbit anti-FAM171A2 | ThermoFisher | Cat#: PA5-71105; RRID: AB_2689922 |
| Rabbit anti-RPTPd | Novus | Cat#: NBP2-49153 |
| Goat anti-RPTPs | R&D Systems | Cat#: AF3430; RRID: AB_2175157 |
| Sheep anti-Teneurin-4 | R&D Systems | Cat#: AF6320; RRID: AB_10920937 |
| Mouse anti-PSA-NCAM1 | Merck Millipore | Cat#: MAB5324; RRID: AB_95211 |
| Rat anti-NCAM2 | R&D Systems | Cat#: MAB778; RRID: AB_2149709 |
| Sheep anti-ISLR2 | R&D Systems | Cat#: AF4650; RRID: AB_2126610 |
| Goat anti-NEGR1/Kilon | R&D Systems | Cat#: AF5394; RRID: AB_2150086 |
| Mouse anti-Neuronal pentraxin 1 | BD Biosciences | Cat#: 610369; RRID: AB_397754 |
| Goat anti-NRP1 | R&D Systems | Cat#: AF566; RRID: AB_355445 |
| Goat anti-NRP2 | R&D Systems | Cat#: AF2215; RRID: AB_2155371 |
| Sheep anti-Noelin | R&D Systems | Cat#: AF4636; RRID: AB_2157225 |
| Goat anti-Plexin-A3 | Thermo-Fisher | Cat#: PA5-47571; RRID: AB_2608296 |
| Rabbit anti-ROBO2 | Aviva Systems | Cat#: ARP45396_P050; RRID: AB_2047840 |
| Goat anti-Neogenin | R&D Systems | Cat#: AF1079; RRID: AB_2151002 |
| Sheep anti-SALM2 | R&D Systems | Cat#: AF5669; RRID: AB_1964706 |
| Mouse anti-Tenascin-R | R&D Systems | Cat#: MAB1624; RRID: AB_2207001 |
| Sheep anti-Contactin 1 | R&D Systems | Cat#: AF7549 |
| Goat anti-CD200 | R&D Systems | Cat#: AF2724 |
| Goat anti-ICAM5 | Novus | Cat#: NB100-53815; RRID: AB_829163 |
| Goat anti-SCFR/Kit | R&D Systems | Cat#: AF1356; RRID: AB_354750 |
| Mouse anti-LSAMP | Developmental Studies Hybridoma Bank | Cat#: 2G9; RRID: AB_2138210 |
| Mouse anti-Trk-C | R&D Systems | Cat#: MAB373; RRID: AB_2155422 |
| Goat anti-Plexin-A1 | R&D systems | Cat#: AF4309; RRID: AB_10645644 |
| Rat anti-HA | Roche | Cat#: 11867423001; RRID: AB 390918 |
| Rabbit anti-pERM | Cell Signaling | Cat#: 3726; RRID:AB 10560513 |
| Goat anti-Chicken Alexa 488 | Thermo-Fisher | Cat# A-11039; RRID: AB_2534096 |

| | | |
|---|---|---|
| Donkey anti-Goat Alexa 488 | Thermo-Fisher | Cat#: A-11055; RRID: AB_2534102 |
| Donkey anti-Goat Alexa 647 | Thermo-Fisher | Cat#: A-21447; RRID: AB_2535864 |
| Donkey anti-Rabbit Alexa 488 | Thermo-Fisher | Cat#: A-21206; RRID: AB_2535792 |
| Donkey anti-Rabbit Alexa 555 | Thermo-Fisher | Cat#: A-31572; RRID: AB_162543 |
| Donkey anti-Rabbit Alexa 647 | Thermo-Fisher | Cat#: A-31573; RRID: AB_2536183 |
| Donkey anti-Mouse Alexa 488 | Thermo-Fisher | Cat#: A21202; RRID: AB_141607 |
| Donkey anti-Mouse Alexa 555 | Thermo-Fisher | Cat#: A31570; RRID: AB_2536180 |
| Donkey anti-Mouse Alexa 647 | Thermo-Fisher | Cat#: A-31571; RRID: AB_162542 |
| Donkey anti-Sheep Alexa 488 | Thermo-Fisher | Cat# A-11015; RRID: AB_2534082 |
| Donkey anti-Sheep Alexa 647 | Thermo-Fisher | Cat#: A-21448; RRID: AB_2535865 |
| Donkey anti-Chicken Alexa 488 | Jackson ImmunoResearch | Cat# 703-545-155; RRID: AB_2340375 |
| Donkey anti-Rat Alexa 647 | Jackson ImmunoResearch | Cat#: 712-605-153; RRID: AB_2340694 |
| Donkey anti-Guinea Pig Cy3 | Jackson ImmunoResearch | Cat# 706-165-148; RRID: AB_2340460 |
| Donkey anti-Guinea Pig Alexa 647 | Jackson ImmunoResearch | Cat#: 706-605-148; RRID: AB_2340476 |
| Donkey anti-Human IgG, Fc Fragment Cy3 | Jackson ImmunoResearch | Cat#: 709-165-098; RRID: AB_2340534 |
| HRP Rabbit anti-Goat IgG | Thermo-Fisher | Cat#: 81-1620; RRID: AB_2534006 |
| HRP Rabbit anti-Sheep IgG | Thermo-Fisher | Cat#: 61-8620; RRID: AB_2533942 |
| HRP Goat anti-Mouse IgG | Thermo-Fisher | Cat#: 62-6520; RRID: AB_2533947 |
| HRP Goat anti-Rabbit IgG | Thermo-Fisher | Cat#: 65-6120; RRID: AB_2533967 |
| Placental alkaline phosphatase monoclonal antibody (8B6.18) | Thermo-Fisher | Cat#: MA5-12694; RRID: AB_10978663 |
| HRP mouse anti-Human IgG1 | Serotec | Cat#: MCA514P |
| *Bacterial and virus strains* | | |
| AAV2/9-hsyn1-mGFP | This paper | N/A |
| AAV2/9-hsyn1-mGFP-T2A-Cre | This paper | N/A |
| AAV2/9-Ef1a-DIO-hChR2(E123T/T159C)-EYFP | [126] | Cat#: Addgene plasmid #35509; RRID: Addgene_35509 |
| *Chemicals, peptides, and recombinant proteins* | | |
| IgSF8-Fc | This paper | N/A |
| IgSF8-His | Sino Biological | Cat#: 51042-M08H |
| Tenascin-R-Fc | This paper | N/A |
| Brevican-Fc | This paper | N/A |
| Neurexin1β-Fc | [127] | N/A |
| Hoechst 33342 | Thermo-Fisher | Cat#: H3570 |
| FM™ 4-64FX, fixable analog | Thermo-Fisher | Cat#: F34653 |
| ChromPure Human IgG, Fc fragment | Jackson Immunoresearch | Cat#: 009-000-008; RRID: AB_2337046 |
| AffiniPure Fab Fragment Donkey anti-Mouse IgG (H + L) | Jackson Immunoresearch | Cat#: 715-007-003; RRID: AB_2307338 |
| FuGENE 6 Transfection Reagent | Promega | Cat#: E2691 |
| AMAXA Nucleofector kit | Lonza | Cat#:VPG-1001 |
| 1-Step Ultra TMB-ELISA HRP | Thermo-Fisher | Cat#: 34028 |
| *Critical commercial assays* | | |
| Myelin-Removal Beads II, human, mouse, rat | Miltenyi Biotec | Cat#: 130-096-733 |
| Mix-n-Stain™ CF™ 488 A Antibody Labeling Kit | Sigma-Aldrich | Cat#: MX488AS100; RRID: AB_10961132 |
| Pierce™ Antibody Clean-up Kit | Thermo-Fisher | Cat#: 44600 |
| Gibson Assembly Cloning Kit | New England Biolabs | Cat#: E5510S |
| *Deposited data* | | |
| Proteomic analysis of sorted MF synaptosomes and P2 synaptosomes | Proteome Exchange | PXD013492 |
| *Experimental models: cell lines* | | |
| Human: HEK293T cells | ATCC | Cat#: CRL-11268 RRID: CVCL_1926 |
| Human: HEK293F suspension adapted | Thermo-Fisher | Cat # R790-07 |
| Mouse: hippocampal primary neurons cultured from embryonic day 18 *Igsf8* conditional knock out mice | This paper | N/A |
| *Experimental models: organisms/strains* | | |
| Mouse: Igsf8 cKO (*B6D2;129S2-Igsf8^tm1.1Osb*) | Riken Institute[64] | Cat#: RBRC05637; RRID: IMSR_RBRC05637 |
| Mouse: Tg(Rbp4-cre)KL100Gsat | GENSAT | Cat#: 031125-UCD; RRID: MMRRC_031125-UCD |
| Mouse: C57BL/6 J | JAX | Cat#: 000664; RRID: IMSR_JAX:000664 |
| *Recombinant DNA* | | |
| Plasmid: FUGW-mCherry | [115] | N/A |
| Plasmid: FUGW-CRE-T2A-mCherry | [115] | N/A |
| Plasmid: FUGW-mGFP-T2A-HA-IgSF8 | This paper | N/A |
| Plasmid: FUGW-GFP-T2A-HA-IgSF8 | This paper | N/A |
| Plasmid: pAAV-hsyn1-mGFP | This paper | N/A |
| Plasmid: pAAV-hsyn1-mGFP-T2A-Cre | This paper | N/A |
| Plasmid: pAAV-Ef1a-DIO-hChR2(E123T/T159C)-EYFP | [126] | Cat#: Addgene plasmid #35509; RRID: Addgene_35509 |
| Plasmid: pBOS-EGFP | [128] | N/A |
| Plasmid: pBOS-myc-LRRTM2 | [127] | N/A |
| Plasmid: pEGFP-N1 | Clontech (TaKaRa) | N/A |
| Plasmid: pEGFP-N1-IgSF8 | This paper | N/A |
| Plasmid: pEGFP-N1-Tenascin-R | This paper | N/A |
| Plasmid: 4F2hc-Fc/AP | This paper | |
| Plasmid: ADAM11-Fc/AP | This paper | |
| Plasmid: ADAM22-Fc/AP | This paper | |
| Plasmid: ADAM23-Fc/AP | This paper | |
| Plasmid: ALCAM-Fc/AP | This paper | |
| Plasmid: APMAP-Fc/AP | This paper | |
| Plasmid: BRINP2-Fc/AP | This paper | |
| Plasmid: CADM4-Fc/AP | This paper | |
| Plasmid: CD200-Fc/AP | This paper | |
| Plasmid: CNTN1-Fc/AP | This paper | |
| Plasmid: CNTNAP2-Fc/AP | This paper | |
| Plasmid: CRTAC1-Fc/AP | This paper | |
| Plasmid: ELFN1-Fc/AP | This paper | |
| Plasmid: ELFN2-Fc/AP | This paper | |
| Plasmid: EPHA4-Fc/AP | This paper | |
| Plasmid: EPHA6-Fc/AP | This paper | |
| Plasmid: EPHA7-Fc/AP | This paper | |
| Plasmid: EPHB2-Fc/AP | This paper | |
| Plasmid: EPHB4-Fc/AP | This paper | |
| Plasmid: FAM171A2-Fc/AP | This paper | |
| Plasmid: GPR37L1-Fc/AP | This paper | |
| Plasmid: HAPLN1-Fc/AP | This paper | |
| Plasmid: HEPACAM-Fc/AP | This paper | |
| Plasmid: ICAM5-Fc/AP | This paper | |
| Plasmid: IgSF8-Fc/AP | This paper | |
| Plasmid: ISLR2-Fc/AP | This paper | |
| Plasmid: KIT/SCFR-Fc/AP | This paper | |
| Plasmid: L1CAM-Fc/AP | This paper | |
| Plasmid: LGI1-Fc/AP | This paper | |
| Plasmid: LINGO1-Fc/AP | This paper | |
| Plasmid: Lphn1-Fc/AP | This paper | |
| Plasmid: Lphn3-Fc/AP | This paper | |
| Plasmid: LRFN1/SALM2-Fc/AP | This paper | |
| Plasmid: LRRC4B/NGL3-Fc/AP | This paper | |
| Plasmid: LSAMP/IgLON3-Fc/AP | This paper | |
| Plasmid: NCAM1-Fc/AP | This paper | |
| Plasmid: NCAM2-Fc/AP | This paper | |
| Plasmid: Nectin3-Fc/AP | This paper | |
| Plasmid: NEGR1/IgLON4-Fc/AP | This paper | |
| Plasmid: Neogenin-Fc/AP | This paper | |
| Plasmid: Neurofascin-Fc/AP | This paper | |
| Plasmid: NLGN1-Fc/AP | This paper | |
| Plasmid: NLGN2-Fc/AP | This paper | |
| Plasmid: NLGN3-Fc/AP | This paper | |
| Plasmid: NPTN-Fc/AP | This paper | |
| Plasmid: NPTX1-Fc/AP | This paper | |
| Plasmid: Noelin-Fc/AP | This paper | |
| Plasmid: NrCAM-Fc/AP | This paper | |
| Plasmid: NRP1-Fc/AP | This paper | |
| Plasmid: NRP2-Fc/AP | This paper | |
| Plasmid: NRXN1α-Fc/AP | This paper | |
| Plasmid: NRXN3α-Fc/AP | This paper | |
| Plasmid: NRXN1β-Fc/AP | This paper | |
| Plasmid: NRXN3β-Fc/AP | This paper | |
| Plasmid: Neurotrimin/IgLON2-Fc/AP | This paper | |
| Plasmid: PlexinA1-Fc/AP | This paper | |
| Plasmid: PlexinA2-Fc/AP | This paper | |
| Plasmid: PlexinA3-Fc/AP | This paper | |
| Plasmid: PlexinA4-Fc/AP | This paper | |
| Plasmid: PlexinB1-Fc/AP | This paper | |
| Plasmid: PlexinC1-Fc/AP | This paper | |
| Plasmid: RPTPα-Fc/AP | This paper | |
| Plasmid: RPTPδ-Fc/AP | This paper | |
| Plasmid: RPTPf/LAR-Fc/AP | This paper | |
| Plasmid: RPTPσ-Fc/AP | This paper | |
| Plasmid: ROBO1-Fc/AP | This paper | |
| Plasmid: ROBO2-Fc/AP | This paper | |
| Plasmid: Shisa6-Fc/AP | This paper | |
| Plasmid: SIRPα-Fc/AP | This paper | |
| Plasmid: Teneurin4-Fc/AP | This paper | |
| Plasmid: TenR-Fc/AP | This paper | |
| Plasmid: TrkC-Fc/AP | This paper | |
| Plasmid: VCAM1-Fc/AP | This paper | |
| Mouse IgSF8 cDNA clone | Origene | MC206708 |
| Mouse Tenascin-R cDNA clone | Source Bioscience | IRCKp5014L0710Q |
| Mouse Brevican cDNA clone | Source Bioscience | IRAVp968B01111D |
| *Software and algorithms* | | |
| Fiji | NIH | https://imagej.net/Welcome; RRID: SCR_002285 |
| Imaris | BITPLANE | http://www.bitplane.com/imaris; RRID: SCR_007370 |
| Microscope Image Browser (MIB) | University of Helsinki | http://mib.helsinki.fi/index.htm |
| GraphPad Prism8 | GraphPad Software, Inc | https://www.graphpad.com/scientific-software/prism/; RRID: SCR_002798 |

**Reporting summary**. Further information on research design is available in the Nature Research Reporting Summary linked to this article.

## Data availability

Proteomic analysis of sorted MF synaptosomes and P2 synaptosomes generated during the study is available in Proteome Exchange: PXD013492: UniProt mouse protein database (downloaded on 03-25-2014; The UniProt Consortium 2015; https://www.uniprot.org/); Panther Classification System using the mouse genome as reference (http://www.pantherdb.org/); PubMed (https://www.ncbi.nlm.nih.gov/pubmed/). Source data are provided with this paper.

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

## Acknowledgements

We thank Franck Polleux, Dietmar Schmucker, Pierre Vanderhaeghen, Anirvan Ghosh, Dan Dascenco, Luís Ribeiro, and Sara Calafate for critical reading of the manuscript, and De Wit lab members for helpful discussion and comments. We thank Pier Andrée Penttila and Christèle Nkama (VIB-KU Leuven FACS Core), the VIB-KU Leuven BioImaging Core, Joris Vandenbempt, and Brenda Luong for experimental help; Nicola Fattorelli for data visualization advice; Etienne Herzog and Matthew Holt for experimental advice. Leica SP8x confocal microscope was provided by InfraMouse (KU Leuven-VIB) through a Hercules type 3 project (ZW09-03). N.A. is supported by the Fundação para a Ciência e a Tecnologia (FCT, Grant Number SFRH/BD/128869/2017). D.C. is supported by NSF IOS Grant #1755189, RWJ Foundation Grant #74260, and Research for Life (WMRF) 2019/301. J.N.S. is supported by R01AG061787. J.d.W is supported by ERC Starting Grant (#311083), FWO Odysseus Grant, FWO Project Grant G0C4518N, FWO EOS Grant G0H2818N, and Methusalem Grant of KU Leuven/Flemish Government.

## Author contributions

N.A., J.N.S., and J.d.W. conceived the study and designed experiments. N.A., S.N.S., J.V., G.C., V.R., J.t.B., L.T., S.P., K.M.V., N.V.G., D.C., and K.D.W. performed experiments and analyzed data. N.A. and J.d.W. wrote the paper with input from all authors. All authors contributed to and approved the final version.

## Competing interests

The authors declare no competing interests.
