## [Peer Review File · Nature Communications]

Reviewers' Comments:

Reviewer #1:

Remarks to the Author:

This study attempts synapse type-specific proteomic profiling using mossy fiber (MF) synapses. The authors find multiple types of cell surface proteins (CSPs), many of which have not been identified. One of them were IgSF8, and the author identify, through unbiased an screen, tenR as a novel binding partner of IgSF8. The authors then characterize mice lacking IgSF8 and reveal that IgSF8 is localized at MF main boutons and filopodia, and find that the loss of IgSF8 leads to a reduction in the number of MF synapses and an increase in the excitability of CA3 neuron by impairing the feedforward inhibition involving Pv-positive interneurons.

This study employ a novel proteomic approach to explore the components of synapse type-specific proteome, and identify a large number of CSPs that will be useful to the community of researchers interested in CA3 or hippocampal neurobiology. The interaction between IgSF8 and tenR is novel and supported by high-quality datasets. Data associated with the structural and functional consequences of IgSF8 deletion are also of high quality and intriguing. I have only the following very minor comments.

1. The manuscript focuses on the distribution of IgSF8 in the hippocampus. How do the distribution patterns of IgSF8 in the whole brain look like? For instance, does IgSF8 show strong region specific or laminar/layer specific patterns in areas including the cortex?
2. Fig. 6a. The frequency of sEPSCs is reduced in CA3 neurons in the IgSF8-mutant mice. However, sEPSC frequency is often influenced by external influences such as network activity or compensation, although EM results seem to support the conclusion that the number of MF boutons is reduced. This reason why the authors did not try mEPSC measurements should be explained.
3. The authors suggest that the images in Fig. 2e show strong laminar distribution of the proteins investigated. However, the image for NCAM2 seems to differ from others and does not seem to support the laminar distribution of the protein.
4. The immunoblots for Fig. 2d need size markers.
5. The binding assays in Fig. 3f ideally need additional negative controls for TenR-Fc and IgSF8-Fc.

Reviewer #2:

Remarks to the Author:

Apostolo and colleagues report a proteomic analysis of mossy fibre synapses and discover many proteins previously not associated with synapses, attesting to the rich complexity of the synapse proteome. Focussing on cell surface proteins (CSPs) the authors identify a set of such proteins, perform immunolocalisation studies and biochemical enrichment experiments. Choosing 73 proteins they perform an in vitro protein-protein interaction between their extracellular domains and identified some novel interaction pairs. They focussed in IgSF8 and its interaction partner TenR, defined in recombinant and pull-down screens. Subcellular localisation studies show IgSF8 suggest a predominantly presynaptic expression. Using IgSF8 knockout mice they show abnormal active zones and filopodia. Perhaps unsurprisingly, the synaptic physiology is abnormal. Finally, the authors study feedforward inhibition mechanisms and show circuit abnormalities.

The authors should be congratulated for a comprehensive analysis with integration of many techniques. Their work highlights the important role of CSPs, their differential expression and how perturbing their expression can results in ultimately complex neural circuit phenotypes.

The discussion was largely a restatement of the results and this detracts from the impact of the paper. The results are well written and the reader can find out all the details there. I found it difficult to work out what was conceptually new as opposed to specific detail about IgSF8.

Seth Grant

Reviewer #3:

Remarks to the Author:

Apóstolo et al. is a very well designed and comprehensive study of the organization of MF-CA3 excitatory and MF-interneuron synapses. The proteomic analysis of isolated synapses from this microcircuit combined with in-depth validation ensures considerable confidence in the study. This confidence is further bolstered by the validation of several proteins known to be expressed at mossy fibers. The authors go on to characterize several known and novel interactions among cell-surface proteins at these synapses. They zeroed onto the IgSf8-TenR complex as being necessary for normal feedforward inhibition in this circuit, which they comprehensively demonstrated using loss-of-function and electrophysiological approaches. The study has many strengths and few weaknesses, and will undoubtedly appeal to a broad readership in synaptic neuroscience.

Concerns that the authors could address

1. In the Introduction, the authors state, "This feedforward inhibition results in net inhibition of CA3 pyramidal neurons, which changes to excitation upon an increase in GC firing frequency due to rapid depression of the MF-interneuron synapse and strong facilitation of the MF-CA3 synapse."

What is the net output comparing MF-interneuron inhibition and MF-CA3 excitation? How does increased GC firing frequency lead to excitation? The authors could clarify these notions more lucidly.

2. The authors should assess the affinity of binding between IgSf8 and TenR and compare it with well-known synaptic pairs.

3. In the interaction matrix between -Fc and -AP tagged proteins, it was not clear how the appropriate concentrations of each protein were arrived at. Please explain.

4. "Several CSPs with a role in MF synapse development, including the secreted glycoprotein C1QL394, Cadherin-9 (CDH9)36, and the G-protein-coupled receptor-like protein GPR15827, were detected in sorted MF synaptosomes (Supplementary Table 1) but did not reach significance."

I would advise the authors to avoid using the phrase "did not reach significance."

5. In the submitted version of the article, Fig3a axes were unreadable. Further, the tables were not presented in an accessible format, and so I could not discern those lists. With regards to those tables, I have given the authors the benefit of the doubt, but I would like to view those Tables, adequately annotated and preferably in excel format.

6. Multiple other synapse pairs have been reported at MF and other synapses. Could the authors expand the Discussion to include how IgSf8-TenR and other well-known synaptic complexes may differentially or redundantly regulate those synapses.

~ Tabrez J. Siddiqui, University of Manitoba

Response to Reviewers "Synapse type-specific proteomic dissection identifies IgSF8 as a hippocampal CA3 microcircuit organizer", Apóstolo et al., submitted manuscript NCOMMS-20-14408

We thank the reviewers and the editor for their time and effort and their constructive feedback on our manuscript. We were very pleased with the reviewers' positive comments on our work. Below, we have outlined our point-by-point response (in black) to the reviewers' comments (in blue). We hope that we were able to convincingly address their comments and believe that these additions to the original submission have further strengthened the conclusions of our study.

Reviewer #1

This study employs a novel proteomic approach to explore the components of synapse type-specific proteome, and identify a large number of CSPs that will be useful to the community of researchers interested in CA3 or hippocampal neurobiology. The interaction between IgSF8 and tenR is novel and supported by high-quality datasets. Data associated with the structural and functional consequences of IgSF8 deletion are also of high quality and intriguing. I have only the following very minor comments.

We thank the reviewer for these kind words and the positive assessment of our study, which is very much appreciated. We have addressed the minor comments as follows:

1. The manuscript focuses on the distribution of IgSF8 in the hippocampus. How do the distribution patterns of IgSF8 in the whole brain look like? For instance, does IgSF8 show strong region specific or laminar/layer specific patterns in areas including the cortex?

We have added whole-brain overview images and several higher-zoom images of IgSF8 protein distribution in the brain to Supplementary Figure 4b. We do not observe strong staining or layer-specific patterns for IgSF8 in cortex. We do find prominent labeling of several fiber tracts such as corpus callosum, the internal capsule and fimbria. We have indicated these IgSF8-positive structures in Supplementary Figure 4b.

2. Fig. 6a. The frequency of sEPSCs is reduced in CA3 neurons in the IgSF8-mutant mice. However, sEPSC frequency is often influenced by external influences such as network activity or compensation, although EM results seem to support the conclusion that the number of MF boutons is reduced. This reason why the authors did not try mEPSC measurements should be explained.

The reviewer is correct that sEPSC frequency is affected by network activity. The reason we analyzed sEPSCs instead of mEPSCs was primarily a practical one, as the use of TTX for mEPSC analysis does not allow the assessment of other parameters of synaptic function, such as evoked synaptic transmission. For this reason, we analyzed sEPSCs in our previous work on the MF synapse (Condomitti et al., Neuron 2018) and used the same approach in this manuscript. As our EM results supported the observations on reduced sEPSC frequency in the *IgSF8* cKO, we did not further pursue mEPSC analysis. An additional consideration to analyze sEPSCs was that we wanted to couple these observations to our analysis of feedforward inhibition in the CA3 microcircuit, which is inherently sensitive to network activity and compensation and had to be performed in the absence of TTX.

3. The authors suggest that the images in Fig. 2e show strong laminar distribution of the proteins investigated. However, the image for NCAM2 seems to differ from others and does not seem to support the laminar distribution of the protein.

We have replaced the NCAM2 image in Figure 2e with a different one that better shows the laminar distribution of NCAM2 in the hippocampus. NCAM2 laminar staining is observed in DG hilus, CA3 stratum lucidum and in CA1 stratum lacunosum moleculare. Unfortunately, the NCAM2 antibody does not produce strong staining under any of the various tissue treatment conditions we tested, despite repeated attempts to improve the signal.

4. The immunoblots for Fig. 2d need size markers

We have corrected this and added molecular weight indications to the immunoblots in Figure 2d. The full-size blots for these proteins with size marker are also shown in Supplementary Figure 2a.

5. The binding assays in Fig. 3f ideally need additional negative controls for TenR-Fc and IgSF8-Fc.

We apologize that this wasn't clear. The negative controls for TenR-Fc and IgSF8-Fc (TenR-Fc or IgSF8-Fc binding to EGFP-expressing HEK293T cells) were shown in the original submission's Supplementary Figure 3d, which also included a positive control (Nrnx1beta(-splice site 4)-Fc binding to LRRTM2-expressing HEK293T cells). We have now included this control data in the main Figure 3f and reorganized this panel to clarify this point.

Reviewer #2

The authors should be congratulated for a comprehensive analysis with integration of many techniques. Their work highlights the important role of CSPs, their differential expression and how perturbing their expression can result in ultimately complex neural circuit phenotypes.

We thank the reviewer for these kind words and the positive assessment of our study, which is very much appreciated.

The discussion was largely a restatement of the results and this detracts from the impact of the paper. The results are well written and the reader can find out all the details there. I found it difficult to work out what was conceptually new as opposed to specific detail about IgSF8.

We thank the reviewer for pointing this out and have considerably revised the Discussion to better address the conceptual advance and novel findings of our study:

- We have highlighted the advantage of isolating a specific type of synapse with well-characterized properties in enabling us to link proteome information to structural and functional features, at the single-synapse level;
- We have highlighted the conceptual advance of our study in providing the first insight into the CSP landscape and cell-surface interactome of a specific excitatory synapse type;
- We have re-ordered the Discussion to better highlight the characterization of novel, uncharacterized CSPs at the MF synapse;
- We have added a discussion on the CSP interactome network analysis and the novel cell-surface interactions we identify at the MF synapse.
- We have removed redundancies with the preceding sections of the manuscript, in particular on the role of filopodia in CA3 microcircuit function.

Reviewer #3

The study has many strengths and few weaknesses, and will undoubtedly appeal to a broad readership in synaptic neuroscience.

We thank the reviewer for these kind words and the positive assessment of our study, which is very much appreciated. We have addressed the reviewer's concerns as follows:

1. In the Introduction, the authors state, "This feedforward inhibition results in net inhibition of CA3 pyramidal neurons, which changes to excitation upon an increase in GC firing frequency due to rapid depression of the MF-interneuron synapse and strong facilitation of the MF-CA3 synapse."

What is the net output comparing MF-interneuron inhibition and MF-CA3 excitation? How does increased GC firing frequency lead to excitation? The authors could clarify these notions more lucidly.

We have adapted this text to clarify this point as follows:

'Each MF bouton contains a large vesicle pool and multiple release sites capable of providing powerful excitatory input to CA3 pyramidal neurons. This robust excitation is controlled by strong feedforward

inhibition of CA3 pyramidal neurons⁸, mediated by filopodia extending from the MF bouton that form excitatory synapses onto interneurons in stratum lucidum (SL)⁹ (Fig. 1a). SL interneurons in turn provide inhibition to CA3 pyramidal neurons. As MF-interneuron synapses are more numerous than MF-CA3 synapses⁹, this results in net inhibition of CA3 pyramidal neurons. The net inhibition changes to excitation upon an increase in GC firing frequency, due to a rapid depression of the MF-interneuron synapse and a strong facilitation of the MF-CA3 synapse^{10,11}.

2. The authors should assess the affinity of binding between IgSF8 and TenR and compare it with well-known synaptic pairs.

We have measured binding between IgSF8 and TenR with biolayer interferometry using various concentrations of purified proteins and determined the affinity of the interaction at $K_D = 1.389 \mu\text{M}$. We have added this data to Figure 3h and i.

The affinity we measure is in the micromolar range. The interaction is therefore not as strong as that of other ligand-receptor pairs, such as LRRTMs-neurexins for example, which is in the nanomolar range. However, affinity measurements can vary, sometimes in orders of magnitude, depending on the method of analysis used (e.g. quantification of cell-surface binding, SPR, BLI, or ITC). We have therefore not added a discussion in which we compare the affinity of the IgSF8-TenR interaction to the affinity of other synaptic complexes. In our view, this would require a systematic assessment of different synaptic complexes along with IgSF8-TenR using the same method of recombinant protein preparation and affinity measurement.

3. In the interaction matrix between –Fc and –AP tagged proteins, it was not clear how the appropriate concentrations of each protein were arrived at. Please explain.

The ELISA-based assay we used does not estimate or measure the affinity of the interaction of each new ligand-receptor pair and is therefore not quantitative. It is done using conditioned media in which the presence of the recombinant protein was verified, but concentrations were not determined. Because of the sensitivity of this assay, low expression levels do not preclude detection of the interaction (Ranaivoson et al 2019, Ozkan et al., 2013, Visser et al., 2015). Although these experiments have been replicated three times, it will be important for other interested research labs to independently validate them and measure an affinity of interaction with other appropriate methods.

We have added the following text to the Discussion (page 20) to better discuss the limitations of this approach:

'As the MF synapse cell-surface interactome screen was limited to those CSPs that reached significance in our proteome analysis, tests only binary interactions, does not take splice variants into account, and does not detect binding affinities weaker than $\pm 10 \mu\text{M}$ ^{25,26,53,56}, the MF synapse CSP interactome will likely be even more complex. Synapse type-specific proteome analysis and cell-surface interactome analysis will be used in future studies to determine to what extent developmental stage and neural activity influence cell-surface composition and interactome of MF synapses.'

4. "Several CSPs with a role in MF synapse development, including the secreted glycoprotein C1QL3⁹⁴, Cadherin-9 (CDH9)³⁶, and the G-protein-coupled receptor-like protein GPR158²⁷, were detected in sorted MF synaptosomes (Supplementary Table 1) but did not reach significance." I would advise the authors to avoid using the phrase "did not reach significance."

We thank the reviewer for pointing this out and have changed this sentence as follows (page 18):

'Several CSPs with an established role in MF synapse development and function, including the postsynaptic receptors EphB2⁹⁴ and EphA4⁹⁵, and the presynaptic adhesion molecule NRXN1⁹⁶, were identified (see Supplementary Table 2 for a complete overview). In addition, the secreted glycoprotein C1QL3⁹⁷, the classic type II cadherin Cadherin-9 (CDH9)³⁶, and the G-protein-coupled receptor-like protein GPR158²⁷ were also detected in sorted MF synaptosomes (Supplementary Table 1).'

5. In the submitted version of the article, Fig3a axes were unreadable. Further, the tables were not presented in an accessible format, and so I could not discern those lists. With regards to those tables, I have

given the authors the benefit of the doubt, but I would like to view those Tables, adequately annotated and preferably in excel format.

We apologize that the labels of the axes of the interactome matrix in Fig. 3a were too small and have included a new version of this graph with increased font size in Fig. 3a. The interactome matrix is also displayed in Supplementary Table 3 (Tab 'Matrix'), which allows the reader to further zoom in.

The supplementary tables were uploaded in excel format, but listed on the journal website in both PDF format (difficult to read) and excel format (original source files). Possibly this last format was missed by the reviewer or the files did not display correctly because of a technical reason. For the revised version of the manuscript, we have again uploaded the original source files in excel format, which should be fully accessible to all readers.

6. Multiple other synapse pairs have been reported at MF and other synapses. Could the authors expand the Discussion to include how IgSf8-TenR and other well-known synaptic complexes may differentially or redundantly regulate those synapses.

We thank the reviewer for this suggestion and have expanded the Discussion to include additional synaptic CSPs at the MF synapse. We have added this to the section mentioned above in our response to Reviewer #3 Point #4 and have added a sentence on differential and redundant roles at the MF synapse (page 18-19):

'Several CSPs with an established role in MF synapse development and function, including the postsynaptic receptors EphB2⁹⁴ and EphA4⁹⁵, and the presynaptic adhesion molecule NRXN1⁹⁶, were identified (see Supplementary Table 2 for a complete overview). In addition, the secreted glycoprotein C1QL3⁹⁷, the classic type II cadherin Cadherin-9 (CDH9)³⁶, and the G-protein-coupled receptor-like protein GPR158²⁷ were also detected in sorted MF synaptosomes (Supplementary Table 1). Interestingly, the loss of function phenotypes of these proteins display similarities, but are rarely identical to one another. Together, these findings suggest that a combinatorial code of CSPs that act in a partially redundant manner defines MF synapse identity. The CSP diversity at the MF synapse may also reflect heterogeneity within this synaptic population. The maturational state of MF synapses varies due to the continuous integration of newborn GCs into the hippocampal circuit⁹⁸. Different histories of synaptic activity may also diversify CSP composition.'

Reviewers' Comments:

Reviewer #1:

Remarks to the Author:

The authors have fully addressed all of my review comments. I do not have any additional comments.

Reviewer #2:

Remarks to the Author:

The authors have addressed my concerns

Reviewer #3:

Remarks to the Author:

The authors have addressed all my concerns, and am happy to support the publication of this study in Nature Communications.